# Minimax Sample Complexity of Graph Neural Networks: Lower Bounds and Structural Effects

**Ahmad Ghasemi & Hossein Pishro-Nik**
Department of Electrical and Computer Engineering
University of Massachusetts
Amherst, MA 01003, USA
{aghasemi,pishro}@umass.edu

## Abstract

Graph Neural Networks (GNNs) achieve strong empirical performance across domains, yet their fundamental statistical behavior remains poorly understood. This paper develops a minimax analysis of ReLU message-passing GNNs with explicit architectural assumptions, in both inductive (graph-level) and transductive (node-level) settings. For arbitrary graphs without structural constraints, we show that the worst-case generalization error scales as $\sqrt{\log d/n}$ with sample size $n$ and input dimension $d$, matching the $1/\sqrt{n}$ behavior of feed-forward networks. Under a spectral–homophily condition combining strong label homophily and bounded spectral expansion, we prove a stronger minimax lower bound of $d/\log n$ for transductive node prediction. We complement these results with a systematic empirical study on three large-scale benchmarks (ogbn_arxiv, ogbn_products_50k, Reddit_50k) and two controlled synthetic datasets representing the worst-case and structured regimes of our theory. All benchmark graphs we study fall in the slow-mixing, bottlenecked regime captured by our spectral–homophily condition, and ratio-based scaling tests show error decay consistent with the $d/\log n$ rate in real and structured settings, while the worst-case synthetic dataset follows the $\sqrt{\log d/n}$ curve. Together, these results indicate that practical GNN tasks often operate in the spectral–homophily regime, where our lower bound $d/\log n$ is tight and effective sample complexity is driven by graph topology rather than universal $1/\sqrt{n}$ behavior.

## 1 Introduction

Graph Neural Networks (GNNs) have become a standard tool for learning from structured data, powering state-of-the-art systems in social networks, molecular property prediction, recommendation, and community analysis (Sen et al., 2008; Ruddigkeit et al., 2012; Ramakrishnan et al., 2014). Their success stems from message passing: node representations are iteratively updated using features from local neighborhoods. Despite this broad empirical impact, the statistical foundations of GNNs remain poorly understood. In particular, a fundamental open question persists: *How many training samples are needed for a GNN to generalize on a given graph, and how does graph structure shape this requirement?*

For classical feed-forward networks, minimax analyses show that ReLU architectures achieve generalization error scaling as $1/\sqrt{n}$ with $n$ i.i.d. samples (Golestaneh et al., 2024), contrasting with the simpler $1/n$ rate in parametric models. GNNs, however, break the independence assumptions underlying these results: node samples are correlated through edges, message passing couples distant regions of the graph, and the effective number of statistically independent observations can differ dramatically from the number of labeled nodes. As a result, the sample complexity of GNNs cannot be inferred from standard deep-learning theory, and must instead reflect the interplay between architecture and graph topology. This raises a central question for modern GNN practice: *What are the minimax limits for GNNs, and under what structural conditions do they arise?*

Most prior theoretical work addresses only the *inductive* graph-level regime, where each training example is an independent graph. In contrast, many widely used benchmarks, including `ogbn_arxiv`, `ogbn_products`, and `Reddit`, operate in the *transductive* node-level setting: a single fixed graph is observed, a subset of nodes is labeled, and the model must generalize across the same graph structure. These two regimes differ sharply in their statistical difficulty. Independent graphs behave like classical samples, whereas node labels collected on a single slowly mixing graph may exhibit substantial redundancy. A principled minimax analysis of both regimes is therefore needed to understand when GNNs follow classical $1/\sqrt{n}$ behavior and when structural properties of the graph impose stricter limits.

This paper develops such an analysis. First, we establish a *worst-case* minimax lower bound for ReLU message-passing GNNs in the inductive setting, showing that no estimator can achieve error better than $\Omega\left(\sqrt{\log d / n}\right)$[1], where $d$ is the input dimension. This rate matches the $1/\sqrt{n}$ behavior of classical deep networks and holds on adversarially chosen graphs, such as path graphs, where minimal connectivity forces message passing to propagate information slowly.

Second, we prove a *sharper*, structure-aware minimax lower bound for the transductive node-level regime. Under a natural *spectral–homophily* condition—requiring strong label homophily together with weak spectral expansion, formalized as a small Laplacian spectral gap $\lambda_2 \le \kappa / \log n$—we show that the effective sample size collapses from $n$ to $\Theta(\log n)$ due to highly overlapping message-passing neighborhoods. In this regime, the minimax risk cannot decay faster than $\Omega(d / \log n)$, a significantly slower rate than $1/\sqrt{n}$. This result reveals that graph topology and mixing geometry, rather than neural architecture alone, can fundamentally constrain the statistical efficiency of GNNs.

Our empirical studies complement these theoretical findings using both controlled synthetic datasets and three large-scale real benchmarks. Synthetic worst-case graphs constructed to instantiate Theorem 1 follow the $\sqrt{\log d / n}$ rate exactly, while synthetic bottlenecked graphs satisfying spectral–homophily follow the $d / \log n$ rate, confirming tightness of both bounds. Crucially, the three benchmark graphs we study satisfy the structural (slow-mixing) spectral–homophily condition required by our transductive lower bound (formalized in Section 3). Ratio-based scaling diagnostics then show that their empirical error curves remain stable when normalized by $d / \log n$, and diverge when normalized by $\sqrt{\log d / n}$, indicating that practical GNN learning problems consistently operate in the structure-limited regime predicted by Theorem 2.

The contributions are as follows.

1. We analyze both inductive (graph-level) and transductive (node-level) prediction settings, providing minimax lower bounds tailored to each regime.
2. For arbitrary graphs without structural assumptions, we prove a lower bound of $\mathcal{R} = \Omega\left(\sqrt{\frac{\log d}{n}}\right)$, matching the classical $1/\sqrt{n}$ rate for ReLU networks.
3. Under the spectral–homophily condition $\lambda_2 \le \kappa / \log n$, we show that the minimax risk tightens to $\mathcal{R} = \Omega\left(\frac{d}{\log n}\right)$, reflecting the collapse of effective sample size on slowly mixing graphs.
4. Using three real benchmarks and two controlled synthetic datasets, we combine structural diagnostics, ratio-based scaling tests, and stress tests to demonstrate that real graphs lie in the structural regime and empirically follow the $d / \log n$ scaling predicted by Theorem 2.
5. Our results show that the effective sample complexity of GNNs is governed not only by architecture but by graph topology—particularly homophily, spectral expansion, and mixing time—highlighting the need for structure-aware generalization theory.

## 2 RELATED WORK

The sample complexity of deep neural networks is well studied. For fully connected and convolutional architectures, the minimax risk is known to scale as $1/\sqrt{n}$, reflecting the higher data requirements of deep learning models compared to classical parametric methods (Golestaneh et al., 2024). Nonparametric regression under smoothness assumptions also yields convergence guarantees (Schmidt-Hieber, 2020), though these results differ substantially from those for modern deep architectures.

---

[1] All logarithms are natural unless stated otherwise.

In contrast, the theoretical understanding of generalization in Graph Neural Networks (GNNs) remains underdeveloped. Early efforts analyzed the VC-dimension of GNNs (Scarselli et al., 2009), but obtained bounds that scale poorly with depth and width. PAC-Bayesian approaches provided stability-based alternatives (Liao et al., 2020), yet sharp sample complexity characterizations are still lacking. Other lines of work investigate representational limits (Garg et al., 2020), or connect graph topology to training dynamics (Oono & Suzuki, 2021; Nikolentzos et al., 2022). However, lower bounds on generalization, critical for understanding statistical limitations, remain scarce.

**Expressivity and generalization of MPNNs.** Franks et al. study message-passing GNNs from an expressivity–learnability perspective, establishing *upper* generalization bounds via VC/covering-number analyses, showing how node individualization and positional encodings boost expressivity while preserving learnability (Franks et al., 2024). Their guarantees depend on architectural size and the chosen individualization scheme. Our work is complementary: we establish *minimax lower bounds* for standard ReLU MPNNs with input-independent local aggregation (Assumption (A1)), exposing how graph structure shapes learnability through the spectral–homophily condition (Theorem 2). In short, Franks et al. (2024) characterize what is achievable (upper bounds), while our results certify the fundamental obstacles that remain even for richer hypothesis classes.

Recently, Pellizzoni et al. analyzed GNNs with node individualization schemes, showing that such modifications reduce sample complexity by enhancing expressivity while controlling VC-dimension and covering numbers (Pellizzoni et al., 2024). Together with (Franks et al., 2024), these works chart the *upper-bound* landscape under expressivity-enhancing augmentations (e.g., individualization or positional encodings). Our focus is orthogonal: we establish *lower* bounds for standard message-passing GNNs without such augmentations, exposing an unavoidable dependence on graph structure.

Our work extends the minimax framework from feedforward networks (Golestaneh et al., 2024) to GNNs with arbitrary graph inputs, without relying on strong smoothness or independence assumptions. By incorporating graph topology directly, we derive intrinsic lower bounds on GNN sample complexity that align closely with empirical trends. Unlike our general bound (Theorem 1), the structure-aware bound (Theorem 2) accommodates adjacency-masked attention by relying on mixing/locality rather than input-independent aggregation.

Taken together, these strands bracket the problem: expressivity-driven *upper* bounds (Pellizzoni et al., 2024; Franks et al., 2024) and structure-aware *lower* bounds (this work).

## 3 PROBLEM FORMULATION AND MAIN RESULT

We consider a GNN operating on a graph $G = (V, E)$ with $|V|$ nodes, $|E|$ edges, adjacency matrix $A$, and node features $X_v \in \mathbb{R}^{|V| \times d}$ for $v \in V$.

**Graphs and terminology.** Throughout, we allow arbitrary simple, undirected graphs. A *chain graph* (path graph $P_m$ on $m$ nodes) has edges $\{(1, 2), (2, 3), \ldots, (m-1, m)\}$. Chain graphs are admissible members of our graph family and instantiate the hard distribution in the proof of Theorem 1.

**Task settings.** We study two prediction regimes with $\hat{Y}$ the output of a GNN $f$, and $q \geq 1$ its output dimension: (i) *Graph-level (inductive)*: Each example is a graph $G$ with features $X$, and the model outputs $f(G, X) = \hat{Y} \in \mathbb{R}^q$. (ii) *Node-level (transductive)*: A single graph $G$ is observed; training/test examples are nodes $v \in V$. The model outputs $f(G, X) = \hat{Y} \in \mathbb{R}^{|V| \times q}$, with the $v$-th row $\hat{y}_v$ predicting node $v$.

Unless stated otherwise, losses are squared error for regression and cross-entropy for classification. Theorem 1 concerns graph-level (inductive) risk, and Theorem 2 node-level (transductive) risk.

**ReLU Graph Neural Networks.** A ReLU-based GNN with $L$ message-passing layers realizes a function $f \colon G \mapsto \hat{Y}$, where $G$ is a graph with node features $X$, and $\hat{Y}$ is the predicted output. Each layer updates hidden node representations as:

$$h_i^{(\ell+1)} = \phi\left(W^{(\ell)} \operatorname{Agg}_{j \in \mathcal{N}(i)} h_j^{(\ell)} + B^{(\ell)} h_i^{(\ell)}\right), \quad \phi(z) = \max\{0, z\}, \quad \ell = 0, \ldots, L-1. \quad (1)$$

Here, $W^{(\ell)} \in \mathbb{R}^{d_{\ell+1} \times d_\ell}$ acts on the aggregated neighbor messages $\operatorname{Agg}_{j \in \mathcal{N}(i)} h_j^{(\ell)}$, and $B^{(\ell)} \in \mathbb{R}^{d_{\ell+1} \times d_\ell}$ is the self–loop mixing matrix applied to $h_i^{(\ell)}$. Additive biases $b^{(\ell)} \in \mathbb{R}^{d_{\ell+1}}$ may be

included but do not affect the minimax bounds. Agg is a permutation-invariant, input-independent aggregator (e.g., sum or mean). Node representations are initialized as $h_i^{(0)} = x_i$.

**Architectural scope and assumptions.** Our lower bound in Theorem 1 applies to message-passing GNNs that satisfy: **(A1)** input-independent, 1-hop permutation-invariant aggregation (e.g., SUM, MEAN, normalized adjacency), and **(A2)** uniform layerwise Lipschitz/variation control, instantiated as the $\ell_1$-norm budget $\sum_{\ell=0}^{L-1} (\|W^{(\ell)}\|_1 + \|B^{(\ell)}\|_1) \leq v_s$, which promotes sparsity and is consistent with recent theoretical results on over-parameterized networks (Lederer, 2022; Taheri et al., 2020). (Any equivalent operator-norm bound yields the same rates up to constants.)

Transformers and attention-based GNNs violate (A1) and are therefore excluded from Theorem 1. By contrast, Theorem 2 requires only adjacency locality and bounded layer operators, and thus extends to adjacency-masked attention under suitable norm bounds (see Remarks 2).

We assume ReLU activations, standard in GCNs, GATs, and GraphSAGE; the minimax bounds also hold for any larger class formed by replacing ReLU with more expressive or injective MLPs.

We define $\mathcal{F}_{\mathrm{GNN}}(v_s, L)$ as the class of $L$-layer ReLU GNNs satisfying this constraint. For simplicity, we fix $(v_s, L)$ and write $\mathcal{F}_{\mathrm{GNN}}$.

**Norms.** We use the following norms throughout. For a matrix $A \in \mathbb{R}^{m \times n}$, the entrywise $\ell_1$ norm is $\|A\|_1 = \sum_{i,j} |A_{ij}|$, used for weight and bias constraints. For a vector $v \in \mathbb{R}^d$, $\|v\|_2$ denotes the Euclidean norm $\|v\|_2 = \left(\sum_{i=1}^d v_i^2\right)^{1/2}$. For a matrix $A$, $\|A\|_2$ denotes the spectral norm (largest singular value). For functions $f : \mathcal{X} \to \mathbb{R}$, the $L_2$ norm under $\mathcal{P}$ is $\|f\|_{L_2} = \left(\mathbb{E}_{(G,X) \sim \mathcal{P}}[f(G,X)^2]\right)^{1/2}$.

**Risk notions.** We quantify generalization error via minimax risks. Here $f^\star \in \mathcal{F}_{\mathrm{GNN}}$ denotes a target function (ground truth), and $\hat{f}$ a learned estimator depending on training data.

For readers less familiar with minimax theory, we provide a short primer explaining the general formulation and its specialization to regression in Appendix A.

*Graph-level (inductive) risk*: Let $(G_i, X_i, Y_i)_{i=1}^n$ be i.i.d. training samples, where each $G_i$ is an independent graph. Define

$$\mathcal{R}_n^{\mathrm{graph}}(\mathcal{F}_{\mathrm{GNN}}) := \inf_{\hat{f}} \sup_{f^\star \in \mathcal{F}_{\mathrm{GNN}}} \mathbb{E}_{\mathrm{train}} \mathbb{E}_{G \sim \mathbb{P}_G} \left[(\hat{f}(G) - f^\star(G))^2\right], \tag{2}$$

where $\mathbb{E}_{\mathrm{train}}$ is over the training graphs $(G_i, X_i, Y_i)_{i=1}^n \sim \mathbb{P}^n$ and the inner expectation is over an independent test graph $G \sim \mathbb{P}_G$.

*Node-level (transductive) risk*: Given a fixed connected graph $G = (V, E)$ with features $X$, let $S \subset V$ be a uniformly random set of $n$ labeled nodes and let $\hat{f} = \hat{f}(\cdot; G, X, S)$ be the learned predictor. We define

$$\mathcal{R}_{(n,G)}^{\mathrm{node}}(\mathcal{F}_{\mathrm{GNN}}) := \inf_{\hat{f}} \sup_{f^\star \in \mathcal{F}_{\mathrm{GNN}}} \mathbb{E}_S \left[\frac{1}{|V|} \sum_{v \in V} (\hat{f}(v) - f^\star(v))^2\right], \tag{3}$$

where the expectation is over the random labeled set $S$. Here, $n$ denotes the number of labeled nodes.

These risks correspond to the inductive (graph-level) and transductive (node-level) settings. We will state explicitly which risk each theorem concerns.

Our first theoretical contribution yields a lower bound on the graph-level (inductive) risk.

**Theorem 1** (Graph-level Minimax Lower Bound (Inductive)). *Let $\mathcal{F}_{\mathrm{GNN}}$ be the class of $L$-layer ReLU GNNs with weights satisfying $\sum_{\ell=0}^{L-1} (\|W^{(\ell)}\|_1 + \|B^{(\ell)}\|_1) \leq v_s$, with $L \geq 1$ and $v_s > 0$. Assume $(G_i, X_i, Y_i)_{i=1}^n$ are i.i.d. samples with $Y_i = f^\star(G_i, X_i) + U_i$, $U_i \overset{\mathrm{i.i.d.}}{\sim} \mathcal{N}(0, \sigma^2)$, $f^\star \in \mathcal{F}_{\mathrm{GNN}}$. Then there exists a constant $K_{new} > 0$ such that, for all $n \geq 1$ and $d \geq 2$,*

$$\mathcal{R}_n^{\mathrm{graph}}(\mathcal{F}_{\mathrm{GNN}}) \geq K_{new} \frac{\sigma v_s}{L} \sqrt{\frac{\log d}{n}}. \tag{4}$$

**Interpretation of Theorem 1.** The risk decays no faster than $1/\sqrt{n}$, matching classical results for fully connected ReLU networks (Golestaneh et al., 2024).

**Sample-size implication.** To guarantee error at most $\epsilon^2$, one must have

$$\epsilon^2 \;\geq\; K_{\text{new}} \frac{\sigma v_s}{L} \sqrt{\frac{\log d}{n}} \;\implies\; n \;\geq\; K_{\text{new}}^2 \frac{\sigma^2 v_s^2}{L^2} \frac{\log d}{\epsilon^4}. \tag{5}$$

Compared to classical finite-dimensional parametric estimators (e.g., linear regression, where $n \geq \sigma^2/\epsilon^2$), GNNs require substantially more data to achieve comparable generalization guarantees.

**Proof Sketch.** We apply Fano's inequality (Fano & Hawkins, 1961) and construct a packing set $\mathcal{M} \subset \mathcal{F}_{\text{GNN}}$ by varying the first-layer weights $W^{(0)}$ on path (chain) graphs. The information-theoretic tools underlying this argument (packing sets, the Varshamov–Gilbert bound, and the KL formula for Gaussian regression) are recalled in Appendix C, while the fixed-radius form of Fano's inequality appears in Appendix D. Exhibiting hardness on one such family suffices to establish a minimax lower bound for the unrestricted graph class. Node features are sampled as $X_i \sim \mathcal{N}(0, I_d)$, and labels follow $Y_i = f^*(G_i, X_i) + U_i$, with $U_i \sim \mathcal{N}(0, \sigma^2)$.

*Packing step.* The bound relies on Lemma 2, which constructs a constant-weight Varshamov–Gilbert code realized by first-layer coordinate selectors and shows

$$\log \mathcal{M}(2\epsilon, \mathcal{F}_{\text{GNN}}, \|\cdot\|_{L_2}) \;\geq\; \frac{C_A v_s^2 \log d}{L^2 \epsilon^2} \tag{6}$$

Applying Fano's inequality with KL divergence bounded by $\text{KL}(P_j \| P_k) \leq \frac{2\epsilon^2}{\sigma^2}$ yields

$$\mathcal{R}_{(n, |V|)} \geq \frac{\epsilon^2}{2} \left( 1 - \frac{2n\epsilon^2/\sigma^2 + \log 2}{C_A v_s^2 \log d / L^2 \epsilon^2} \right). \tag{7}$$

Optimizing over $\epsilon^2$ gives the desired bound. The complete proof is provided in Appendix E.

**Remark 1** (Worst-case graphs). *Theorem 1 is established on* path graphs *(chain graphs), where each node has degree at most two. This minimal connectivity creates bottlenecks that slow message passing, making depth the dominant factor. Path graphs thus serve as canonical* worst-case instances*: hardness on this sparse structure certifies the lower bound for all admissible graphs. Although denser graphs may empirically converge faster, the path graph ensures the universal worst-case rate.*

**Remark 2** (Exclusion of attention in Theorem 1). *The packing construction for Theorem 1 exploits assumption (A1), i.e., input-independent local aggregation. Architectures with attention violate (A1) because their mixing weights depend on hidden features; hence the theorem does not apply to graph transformers or attention-based GNNs. This does not contradict the lower bound: by monotonicity of minimax risk, enlarging the hypothesis class cannot reduce the bound.*

Theorem 1 establishes $\sqrt{\frac{\log d}{n}}$ scaling, whereas our empirical results (Section 4) indicate $1/\log n$ scaling in practice. This motivates a refined lower bound under structural graph assumptions, formalized in Theorem 2. We first define the notion of *Spectral–homophily* used therein.

**Spectral–homophily.** Let $\mathcal{L} = I - D^{-1/2} A D^{-1/2}$ be the normalized Laplacian of the graph under consideration and let $\lambda_2(\mathcal{L})$ be its second-smallest eigenvalue. We assume $\lambda_2(\mathcal{L}) \leq \kappa/\log n$ for a constant $\kappa > 0$. This is distinct from label-homophily assumptions (see Appendix G).

**Spectral gap, homophily, and bottleneckedness.** A small $\lambda_2(\mathcal{L})$ indicates weak expansion and slow mixing (e.g., via Cheeger-type relations), creating bottleneck-sparse cuts separating dense communities-that limit how much independent label information message passing can extract. Such structure prevents information injected in one region from propagating globally, as strong homophily (nodes tightly connected within communities) and weak expansion (few inter-community edges) cause messages to "get stuck" within communities. The condition $\lambda_2 \leq \kappa/\log n$ captures this effect: the smaller the gap, the fewer effectively independent samples a GNN receives. Thus, spectral–homophily quantifies the bottleneckedness underlying the $\Omega(d/\log n)$ lower bound in Theorem 2.

**Why the transductive setting amplifies this difficulty.** In the node-level transductive regime, all node features are observed but only a subset of labels. When the graph mixes slowly, these labeled nodes become highly correlated: message-passing neighborhoods overlap, and nearby labels offer

nearly redundant information. Consequently, the setting provides far fewer *effectively independent* signals than the raw label count suggests—only about one in every $O(\log n)$ labels contributes genuinely new information. This reduction in independence, driven by the interaction between message passing and slow graph mixing (rather than the number of labels alone), underlies the $\Theta(\log n)$ effective sample size and yields the $\Omega(d/\log n)$ minimax rate in Theorem 2.

Together, these observations motivate a fundamentally different minimax regime for node-level prediction. When structural bottlenecks force mixing to occur over $\Theta(\log n)$ steps, the $n$ labeled nodes provide far fewer than $n$ effectively independent constraints. Theorem 2 formalizes this intuition by showing that under spectral–homophily, every estimator—regardless of architecture or training procedure—faces a minimax barrier that decays only as $d/\log n$.

**Theorem 2** (Structured-Graph Minimax Lower Bound (Node-Level, Transductive))**.** *Let $L \geq 1$, $v_s > 0$, and let $G = (V, E)$ satisfy the* spectral–homophily *condition $\lambda_2(\mathcal{L}) \leq \kappa/\log n$ for some universal $\kappa > 0$, where $n$ is the number of labeled training nodes and $\mathcal{L}$ is the normalized Laplacian. Then there exists a universal constant $\Gamma > 0$ such that*

$$\mathcal{R}^{\text{node}}_{(n,G)}(\mathcal{F}_{\text{GNN}}) \;\geq\; \frac{\sigma^2 v_s^2}{\Gamma L^2} \cdot \frac{d}{\log n}. \tag{8}$$

As discussed in Appendix G, and formally shown in Lemma 3, the structural condition $\lambda_2 \leq \kappa/\log n$ is asymptotically non-vacuous and excludes all graph families with nonvanishing spectral gap.

**Interpretation of Theorem 2.** This bound decays more slowly than $1/\sqrt{n}$, making it tighter whenever the spectral–homophily condition holds. Extensions to adjacency-masked attention (e.g., GAT) are discussed in Appendices I–J, and practical guidance on improving constants without changing the $\Omega(d/\log n)$ rate is in Appendix K. If spectral–homophily condition fails (e.g., $\lambda_2$ is larger, indicating strong expansion), the independence argument breaks down and the analysis reverts to Theorem 1, yielding the $\Omega(\sqrt{\log d/n})$ rate.

**Sample-size implication.** To achieve generalization error $\epsilon^2$, the following must hold:

$$\frac{\sigma^2 v_s^2}{\Gamma L^2} \frac{d}{\log n} \;\leq\; \epsilon^2 \quad \Longrightarrow \quad n \;\geq\; \exp\!\left(\frac{\sigma^2 v_s^2 d}{\Gamma L^2 \epsilon^2}\right), \tag{9}$$

implying exponential sample complexity in $1/\epsilon^2$, far worse than polynomial rates.

**Proof Sketch.** The proof formalizes the idea that under spectral–homophily ($\lambda_2 \leq \kappa/\log n$), the $n$ labeled nodes do not act as $n$ independent samples. Slow mixing causes message-passing neighborhoods to overlap heavily, making nearby labels largely redundant. Consequently, the number of statistically independent labels collapses to $K = \Theta(\log n)$.

The argument proceeds by identifying $K$ well-separated nodes whose receptive fields interact only weakly under the slow-mixing condition. On these $K$ nodes, we construct a packing set of GNN functions using constant-weight codewords, ensuring that functions differ in one "block" yet remain within the allowed $\ell_1$ norm budget. Two functions that differ in one block achieve separation of order $(v_s/LK)^2 \Delta$, while the Gaussian noise model keeps the KL divergence between their induced distributions of order $1/K$. Applying the fixed-radius version of Fano's inequality to this $K$-block packing yields a minimax risk lower bound proportional to $d/K = d/\log n$. Thus the slow-mixing structure limits the amount of independent information available to any algorithm, leading to the stated $\Omega(d/\log n)$ rate. The complete construction and technical details are given in Appendix F.

## 4 EMPIRICAL STUDIES

In this section, we present proof-of-concept experiments illustrating how the minimax bounds appear in practice. We evaluate three real benchmark datasets, **ogbn_arxiv**, **ogbn_products_50k**, and **Reddit_50k**, alongside two synthetic settings designed to isolate the behaviors predicted by our theory. The first, **Synthetic-FanoWorstCase (Thm-1)**, directly instantiates the worst-case error curve $\sqrt{\log d/n}$ from Theorem 1. The second, **WorstCase_Bottleneck_20k (Thm-2)**, is a controlled community-bottleneck graph dataset satisfying the spectral–homophily condition $\lambda_2 \leq \kappa/\log n$.

Experiments use three representative GNNs: GCN (Kipf & Welling, 2017), GAT (Veličković et al., 2018), and GraphSAGE (Hamilton et al., 2017a). Full dataset descriptions, preprocessing, and

| Dataset | Spectral Gap ($\lambda_2$) | $\kappa_0$ | Homophily |
|---|---|---|---|
| ogbn_arxiv | 0.2112 | 2.5428 | 0.6551 |
| ogbn_products_50k | 0.9201 | 9.9557 | 0.7956 |
| Reddit_50k | 0.9683 | 10.4769 | 0.7748 |
| WorstCase_Bottleneck_20k | 1.0359 | 10.2586 | 0.3164 |

Table 1: Graph Structural Properties Relevant to Theorem 2

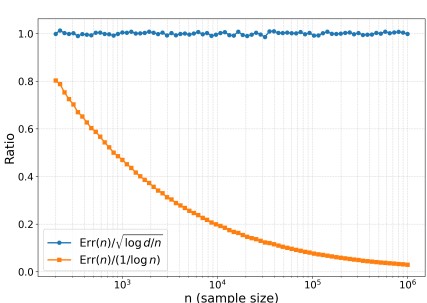

Figure 1: Stability comparison of scaling-law ratios for `Synthetic-FanoWorstCase` (Thm-1).

licensing appear in Appendix M. Details of the synthetic constructions for Theorems 1 and 2 are provided in Appendices O and P, respectively.

**Methodology.** All experiments were implemented in PyTorch Geometric using a unified protocol across datasets. For each dataset, we trained GCN, GAT, and GraphSAGE on a log-spaced grid of sample sizes $n \in \{49, \dots, n_{\max}\}$, where $n_{\max}$ is the size of the training pool: 169,343 for `ogbn_arxiv`, 50,000 for `ogbn_products_50k` and `Reddit_50k`, and 20,000 for the synthetic settings (`Synthetic-FanoWorstCase` (Thm-1) and `WorstCase_Bottleneck_20k`). For each $n$, models were trained under 20 independent seeds (random initialization and random subsampling). To compare empirical behavior with Theorems 1 and 2, we computed the theory-aligned diagnostics $\text{Err}(n)/\sqrt{\log d/n}$ and $\text{Err}(n)/(d/\log n)$. We then aggregated test losses across seeds and fit four candidate scaling laws, $c_1 + \frac{\alpha}{\sqrt{n}}$, $c_2 + \frac{\beta}{n}$, $c_3 + \frac{\delta}{\log n}$, and $c_4 + n^{-\gamma}$, using nonlinear least squares (`curve_fit`) with inverse-variance weighting. Fit quality was evaluated via residual sum of squares (RSS), mean squared error (MSE), $R^2$, and log–log slopes of the error curves.

**Structural Verification of Theorem 2 Conditions.** To test whether each dataset falls in the structural regime of Theorem 2, we compute $\lambda_2(\mathcal{L})$, homophily, and a *single* dataset-level constant $\kappa_0$ that certifies the condition *uniformly* over the training-size grid $\mathcal{N}$. We set $\kappa_0 := \max_{n \in \mathcal{N}} \lambda_2(\mathcal{L}) \log n$ and verify $\lambda_2(\mathcal{L}) \leq \kappa_0/\log n$ for all $n \in \mathcal{N}$. Table 1 reports $\lambda_2(\mathcal{L})$, $\kappa_0$, and homophily, confirming that *real-world graphs lie in the regime where Theorem 2's $d/\log n$ bound applies*. `WorstCase_Bottleneck_20k` satisfies the inequality tightly by construction, while `Synthetic-FanoWorstCase` violates it, yielding a genuine Theorem-1-type worst case. These trends match later results: more homophilic graphs with small spectral gaps mix more slowly, consistent with Theorem 2's $d/\log n$ convergence. Details for computing $\lambda_2(\mathcal{L})$ are in Appendix N.

**Direct Scaling Diagnostics via Error–Ratio Plots (Primary Evidence).** We treat *ratio diagnostics* as the primary empirical test of our theoretical claims. For each dataset–model pair, we compute $\text{Ratio}_1(n) = \text{Err}(n) / \sqrt{\log d/n}$ (Theorem 1 form) and $\text{Ratio}_2(n) = \text{Err}(n) / (d/\log n)$ (Theorem 2 form). A ratio that remains approximately constant across $n$ indicates empirical consistency with the corresponding theoretical rate.

*Synthetic-FanoWorstCase (Thm-1).* As expected, Figure 1 shows that $\text{Ratio}_1(n)$ stays essentially constant and near one, confirming that the synthetic construction follows $\sqrt{\log d/n}$. In contrast, $\text{Ratio}_2(n)$ decreases steadily with $n$, indicating that the $d/\log n$ scaling does *not* fit the Theorem-1 instance. This behavior verifies the correctness of the construction. Additional controlled tests isolating the $n^{-1/2}$ and $\sqrt{\log d}$ dependencies appear in Appendix R.

*Real-World Datasets.* Figures 2, 3, and 4 show that across all three datasets and architectures, $\text{Ratio}_2(n) = \text{Err}(n)/(d/\log n)$ stays nearly flat over two to three orders of magnitude in $n$, while $\text{Ratio}_1(n) = \text{Err}(n)/\sqrt{\log d/n}$ increases steadily, often sharply. This highlights a clear pattern: **real GNN datasets empirically follow the $d/\log n$ scaling predicted by Theorem 2.**

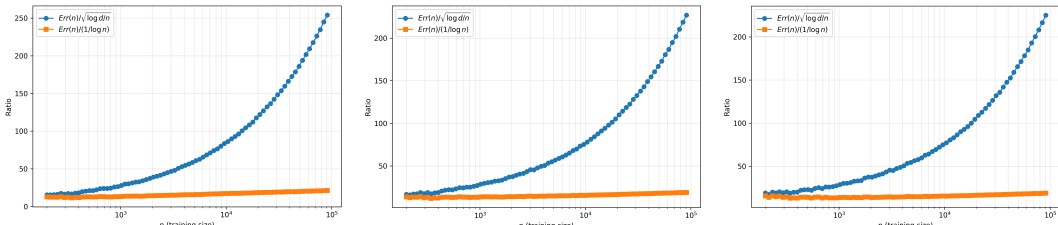

Figure 2: Stability comparison of scaling-law ratios for ogbn_arxiv (left: GAT, middle: GCN, right: GraphSAGE).

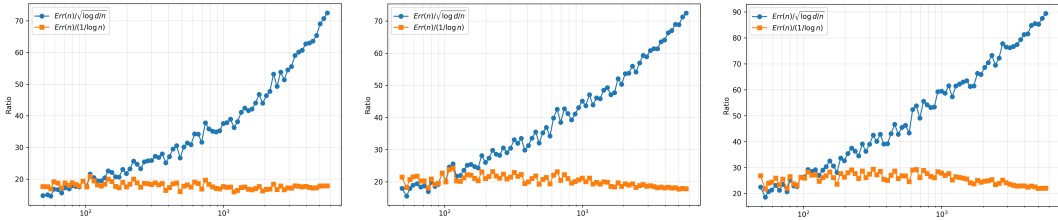

Figure 3: Stability comparison of scaling-law ratios for ogbn_products_50k (left: GAT, middle: GCN, right: GraphSAGE).

**Stress-Testing the Bounds with Synthetic Worst-Case Graphs.** To demonstrate that both minimax bounds are tight in their respective structural regimes, we evaluate the synthetic graph satisfying Theorem 2 assumptions: `WorstCase_Bottleneck_20k`. As shown in Figure 5, $\text{Ratio}_2(n)$ remains stable across $n$ while $\text{Ratio}_1(n)$ increases sharply, mirroring the behavior observed in real datasets. This confirms that the $d/\log n$ rate is tight under the spectral–homophily structure.

**Estimating the Empirical Constant $C^\star$.** To further quantify the tightness of the minimax lower bounds, we estimate the empirical constant $C^\star$ associated with the structured-graph rate. For each dataset and architecture, we compute $C^\star \approx \frac{\text{Err}(n)}{d/\log n}$, the plateau value of the ratio diagnostic $\text{Err}(n)/(d/\log n)$. Across real datasets `ogbn_arxiv`, `ogbn_products_50k`, `Reddit_50k`, $C^\star$ remains stable over several orders of magnitude in $n$, with dataset-specific ranges: approximately 15–25 for `ogbn_arxiv`, 18–22 for `ogbn_products_50k`, and 10–20 for `Reddit_50k`. For the synthetic `WorstCase_Bottleneck_20k` benchmark, $C^\star$ is in the range 8–12, consistent with its sharper bottleneck structure. This stability supports the conclusion that the empirical error scales proportionally to $d/\log n$ within a controlled constant factor, as predicted by Theorem 2.

**Supplementary Curve-Fit Analysis.** Curve fitting is only secondary evidence in our empirical study, since fits can conflate noise, architecture bias, and optimization variance and thus, they are not a reliable basis for testing minimax rates. In our experiments, $1/\log n$ is the best-fit model in only (3/13) architecture–dataset combinations, so curve fits alone do *not* reliably reveal the scaling law. For transparency, Appendix Q shows curve-fit plots for `ogbn_arxiv` and `Reddit_50k` and provides complete raw-error tables (mean $\pm$ standard deviation over seeds) for each dataset, architecture, and training size; Table 2 reports the corresponding fit metrics.

| Dataset | Model | $c_1 + \frac{a}{\sqrt{n}}$ | | | $c_2 + \frac{b}{n}$ | | | $c_3 + \frac{\delta}{\log n}$ | | | $c_4 + \frac{1}{n^\gamma}$ | | | Best Fit |
|---|---|---|---|---|---|---|---|---|---|---|---|---|---|---|
| | | RSS | MSE | $R^2$ | RSS | MSE | $R^2$ | RSS | MSE | $R^2$ | RSS | MSE | $R^2$ | |
| Synthetic | FanoWorstCase | 1.9208e-04 | 2.4010e-06 | 0.9984 | 1.1953e-02 | 1.4941e-04 | 0.9022 | 6.5175e-03 | 8.1469e-05 | 0.9467 | 3.5766e-03 | 4.4707e-05 | 0.9707 | $1/\sqrt{n}$ |
| ogbn_arxiv | GAT | 2.2867e-01 | 2.8584e-03 | 0.8103 | 9.6304e-02 | 1.2038e-03 | 0.9201 | 3.6116e-01 | 4.5145e-03 | 0.7004 | 4.6677e-01 | 5.8347e-03 | 0.6128 | $1/n$ |
| ogbn_arxiv | GCN | 1.7595e-01 | 2.1993e-03 | 0.9589 | 2.7996e-01 | 3.4995e-03 | 0.9345 | 4.0788e-01 | 5.0985e-03 | 0.9046 | 2.0546e+00 | 2.5683e-02 | 0.5195 | $1/\sqrt{n}$ |
| ogbn_arxiv | GraphSAGE | 4.5049e-01 | 5.6311e-03 | 0.9437 | 3.0781e-01 | 3.8476e-03 | 0.9615 | 1.0570e+00 | 1.3212e-02 | 0.8678 | 4.8763e+00 | 6.0954e-02 | 0.3903 | $1/n$ |
| ogbn_products_50k | GAT | 2.5313e+00 | 3.1641e-02 | 0.9493 | 8.2054e+00 | 1.0257e-01 | 0.8357 | 1.9013e+00 | 2.3767e-02 | 0.9619 | 4.0216e+01 | 5.0270e-01 | 0.1946 | $1/\log n$ |
| ogbn_products_50k | GCN | 6.9206e+00 | 8.6508e-02 | 0.9042 | 1.7162e+01 | 2.1452e-01 | 0.7625 | 4.7490e+00 | 5.9363e-02 | 0.9343 | 6.0513e+01 | 7.5642e-01 | 0.1626 | $1/\log n$ |
| ogbn_products_50k | GraphSAGE | 1.3516e+01 | 1.6895e-01 | 0.8577 | 2.8754e+01 | 3.5942e-01 | 0.6972 | 9.4352e+00 | 1.1794e-01 | 0.9006 | 8.1544e+01 | 1.0193e+00 | 0.1413 | $1/\log n$ |
| Reddit_50k | GAT | 2.3900e-01 | 3.9833e-03 | 0.9354 | 3.5081e-01 | 5.8468e-03 | 0.9052 | 3.3451e-01 | 5.7418e-03 | 0.9069 | 2.0218e+00 | 3.3697e-02 | 0.4537 | $1/\sqrt{n}$ |
| Reddit_50k | GCN | 8.0815e-01 | 1.0102e-02 | 0.8610 | 3.3071e-01 | 4.1339e-03 | 0.9431 | 1.3027e+00 | 1.6283e-02 | 0.7759 | 3.6335e+00 | 4.5419e-02 | 0.3749 | $1/n$ |
| Reddit_50k | GraphSAGE | 3.8742e+00 | 4.8427e-02 | 0.8522 | 1.6209e+00 | 2.0261e-02 | 0.9382 | 6.1461e+00 | 7.6827e-02 | 0.7655 | 2.1175e+01 | 2.6469e-01 | 0.1922 | $1/n$ |
| WorstCase_Bottleneck_20k | GAT | 1.5671e+00 | 2.6119e-02 | 0.9177 | 2.1349e-01 | 3.5581e-03 | 0.9888 | 2.6996e+00 | 4.4993e-02 | 0.8582 | 1.6011e+01 | 2.6685e-01 | 0.1587 | $1/n$ |
| WorstCase_Bottleneck_20k | GCN | 2.5240e-01 | 4.2067e-03 | 0.9571 | 4.8768e-02 | 8.1279e-04 | 0.9917 | 5.3007e-01 | 8.8346e-03 | 0.9099 | 4.2115e+00 | 7.0192e-02 | 0.2838 | $1/n$ |
| WorstCase_Bottleneck_20k | GraphSAGE | 9.3697e-02 | 1.5616e-03 | 0.9927 | 4.6429e-01 | 7.7382e-03 | 0.9638 | 3.3031e-01 | 5.5052e-03 | 0.9742 | 1.0156e+01 | 1.6927e-01 | 0.2079 | $1/\sqrt{n}$ |

Table 2: Comparison of Fit Metrics Across All Models and Datasets (Updated Results)

Unlike the ratio diagnostics, which unambiguously favor Theorem 2, the curve fits show mixed behavior: on `ogbn_products_50k` the best fits tend to favor $1/\log n$, whereas on `ogbn_arxiv` and `Reddit-50k` the fits sometimes prefer $1/n$ or $1/\sqrt{n}$. This is expected and does not contradict theory: Curve-fit comparisons reflect finite-sample interpolation accuracy, not asymptotic minimax

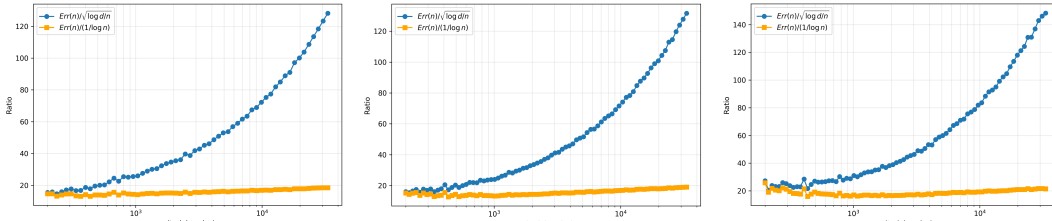

Figure 4: Stability comparison of scaling-law ratios for Reddit_50k (left: GAT, middle: GCN, right: GraphSAGE).

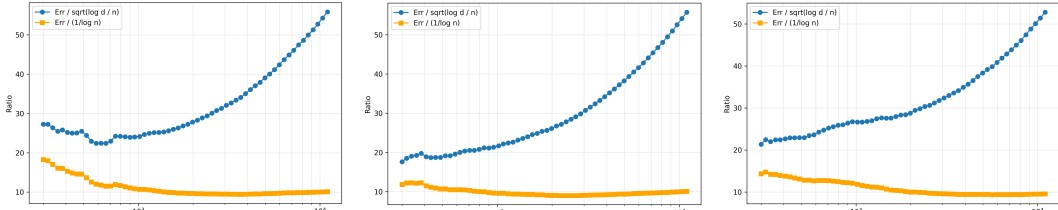

Figure 5: Stability comparison of scaling-law ratios for WorstCase_Bottleneck_20k (left: GAT, middle: GCN, right: GraphSAGE).

behavior. Ratio diagnostics directly test asymptotic structure and therefore carry higher evidential weight. Thus, curve fits serve as useful supporting evidence but are not the primary validation method.

**Summary.** By integrating structural verification, ratio-based scaling diagnostics, synthetic stress tests, curve fits, and raw-result tables, our empirical analysis consistently reveals that: (1) all real-world graph datasets lie inside the structural regime required for Theorem 2; (2) ratio diagnostics unambiguously select the $d/\log n$ rate over $\sqrt{\log d/n}$ across architectures and datasets; (3) synthetic graphs constructed to satisfy Theorem 2 behave like real datasets, while the Theorem-1 synthetic graph behaves in the opposite way; and (4) curve fits do not contradict this conclusion.

*Across real benchmarks with weak spectral gaps and moderate-to-high homophily, empirical convergence often decays far more slowly than the classical $1/\sqrt{n}$ rate, frequently approaching $1/\log n$. This slower decay matches the structural constraints captured by Theorem 2, indicating that GNNs may need larger training sets to generalize reliably on graphs with long mixing times or bottlenecked communities. These trends highlight the importance of structure-aware generalization theory: effective rates depend on graph topology and mixing geometry rather than universal assumptions.*

**Taken together, the evidence shows that practical GNN learning problems operate in a spectral–homophily regime where Theorem 2 provides the correct characterization of sample complexity.**

## 5  CONCLUSION

This paper establishes the first minimax characterization of GNN sample complexity across both inductive and transductive regimes. We show that the familiar $\sqrt{\log d/n}$ rate arises only in adversarial graph settings, while realistic graphs with slow mixing and strong community bottlenecks obey a fundamentally harder limit: a structure-driven $\Omega(d/\log n)$ minimax rate. This reveals that graph topology, not architecture, dictates the effective sample size available to GNNs.

Our empirical results deliver a clear message. Theorem–1 synthetic graphs follow the $\sqrt{\log d/n}$ curve exactly, but all real benchmarks and the Theorem–2 synthetic construction show stable $d/\log n$ behavior across architectures and multiple orders of magnitude in $n$. Structural diagnostics further confirm that real graphs lie squarely within the spectral–homophily regime where Theorem 2 is tight.

These findings overturn the assumption that GNNs inherit classical $1/\sqrt{n}$–type generalization and instead demonstrate that practical GNN learning is typically structure-limited. Future work should develop architectures, sampling schemes, or pre-training strategies that counteract slow mixing, and extend structure-aware analyses to attention-based and higher-order models. Our results chart a clearer theoretical roadmap: generalization on graphs is governed by mixing geometry, and any scalable GNN methodology must contend with this structural barrier.

## 6 REPRODUCIBILITY STATEMENT

We have taken several steps to ensure the reproducibility of our work. All theoretical assumptions, theorems, and the proof sketch of Theorems 1 and 2 are explicitly stated in Section 3. The complete proofs of Theorems 1 and 2 are provided in Appendix E and Appendix F, respectively. Supporting technical components, including primer on minimax risk (Appendix A), degenerate GNN realizations (Appendix B), information-theoretic tools (Appendix C), Fano's inequality (Appendix D), spectral and homophily assumptions (Appendix G), operator-norm control for attention (Appendices I–J), synthetic worst-case construction of Theorem 1 (Appendix O), and synthetic structured bottleneck dataset for Theorem 2 (Appendix P) are all provided for completeness. Experimental protocols are described in Section 4, while dataset descriptions, training procedures, and infrastructure details appear in Appendix M. To further support verification, we provide the full source code as supplementary material, including implementations for data loading, model training, evaluation, and error analysis. The package also contains scripts to reproduce all experimental results, regenerate LaTeX tables, and visualize learning curves. Together, these resources ensure that both the theoretical and empirical results reported in this paper can be independently reproduced and validated.

## 7 ACKNOWLEDGMENT

This work was supported by NSF under Grant CNS-2150832 and Grant CNS-2528914.

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

APPENDIX CONTENTS

## A  PRIMER ON MINIMAX RISK AND REGRESSION AS A SPECIAL CASE

Minimax theory provides a principled way to quantify the best achievable performance of any estimator over a given hypothesis class. Because this background may be unfamiliar to some readers in the GNN community, we provide a short overview. A more complete treatment can be found in Chapter 15 of (Wainwright, 2019b).

## A.1 GENERAL MINIMAX FORMULATION

Let $\Theta$ denote a parameter or function class, and let $\mathcal{P}_\theta$ be the distribution of observed data under parameter $\theta \in \Theta$. For a loss function $L(\widehat{\theta}, \theta)$, the *minimax risk* is defined as

$$\mathfrak{M}_n(\Theta) \;=\; \inf_{\widehat{\theta}} \sup_{\theta \in \Theta} \mathbb{E}_{\mathcal{P}_\theta}\left[L(\widehat{\theta}, \theta)\right]. \tag{10}$$

This quantity characterizes the *best* estimator (infimum over all measurable procedures) against the *worst-case* parameter in $\Theta$ (supremum over $\theta$). A lower bound on $\mathfrak{M}_n(\Theta)$ therefore shows that *no algorithm* can achieve error smaller than this rate.

## A.2 REGRESSION AS A SPECIAL CASE

The supervised regression problem studied in this paper is an instance of the minimax framework. We assume the labels follow the model

$$Y = f^\star(G, X) + U, \qquad U \sim \mathcal{N}(0, \sigma^2), \tag{11}$$

where $f^\star$ is the true target function. We restrict $f^\star$ to the hypothesis class $\mathcal{F}_{\mathrm{GNN}}$, consisting of $L$-layer ReLU GNNs with width $d$.

The loss function is mean squared error:

$$L(f, f^\star) \;=\; \mathbb{E}\left[(f(G, X) - f^\star(G, X))^2\right]. \tag{12}$$

The minimax risk for the regression setting is therefore

$$\mathfrak{M}_n(\mathcal{F}_{\mathrm{GNN}}) \;=\; \inf_{\widehat{f}} \sup_{f^\star \in \mathcal{F}_{\mathrm{GNN}}} \mathbb{E}\left[\left(\widehat{f}(G, X) - f^\star(G, X)\right)^2\right]. \tag{13}$$

This is the precise quantity lower-bounded in Theorem 1 (inductive graph-level setting) and Theorem 2 (transductive node-level setting). In our analysis, the sample size $n$ corresponds to:

- **Inductive (graph-level):** the number of i.i.d. labeled graphs.
- **Transductive (node-level):** the number of labeled nodes in a single graph.

The statistical difficulty differs across these two regimes due to independence in the inductive case versus strong dependence induced by graph connectivity in the transductive case.

## B  DEGENERATE GNN REALIZATION

We construct a one-layer ReLU GNN on the original graph (with self-loops) using the identity aggregator, $\mathrm{Agg} = \mathrm{identity}$. In this case, each node aggregates only its own features—a degenerate but still admissible instance of the message passing. With weights set as $W_j = \frac{v_s}{LK} s_j^{(\ell)}$ and zero bias, the network output is

$$f_{\boldsymbol{s}}(x) = \sum_{\ell=1}^{K} \frac{v_s}{LK} \sum_{j=1}^{d} s_j^{(\ell)} \, \phi(x_j),$$

which lies in $\mathcal{F}_{\mathrm{GNN}}(v_s, 1)$. Although message passing here reduces to self-loops, this subclass is included in our hypothesis space. Since minimax lower bounds apply to any subclass, establishing hardness for these degenerate cases certifies hardness for the full class.

## C  INFORMATION-THEORETIC TOOLS

For completeness, we record the remaining tools used in the proofs of Theorems 1 and 2. These results are standard in empirical-process and information-theoretic lower bounds.

**Packing sets and packing number.** Let $(\mathcal{F}, \rho)$ be a metric space. A set $\{f_1, \ldots, f_M\} \subset \mathcal{F}$ is a $\delta$-*packing* if $\rho(f_i, f_j) \geq \delta$ for every $i \neq j$. The *packing number* is

$$\mathcal{M}(\delta, \mathcal{F}, \rho) := \sup\{M : \exists \text{ a } \delta\text{-packing of size } M\}.$$

This definition follows (Wainwright, 2019a, Def. 5.3).

**Varshamov–Gilbert bound.** There exists a subset $\mathcal{C} \subset \{0, 1\}^d$ with pairwise Hamming distance at least $d/4$ and cardinality

$$|\mathcal{C}| \geq 2^{d/8},$$

as established in (Gilbert, 1952; Varshamov, 1957). This result is used to construct large packing sets for the function classes considered.

**KL divergence for Gaussian regression.** Consider the regression model with fixed design $X_1, \ldots, X_n$ and i.i.d. noise $Y_i = f(X_i) + U_i$, $U_i \sim \mathcal{N}(0, \sigma^2)$ independent. Let $P_f^{(n)}$ denote the joint law of $(Y_1, \ldots, Y_n)$ (conditional on the design). Then

$$\mathrm{KL}(P_f^{(n)} \| P_g^{(n)}) = \frac{1}{2\sigma^2} \sum_{i=1}^{n} (f(X_i) - g(X_i))^2.$$

If the design is random i.i.d. from $P_X$, taking expectation over $X$ yields

$$\mathbb{E}_X\big[\mathrm{KL}(P_f^{(n)} \| P_g^{(n)})\big] = \frac{n}{2\sigma^2} \|f - g\|_{L_2(P_X)}^2.$$

**Use in the proofs.** These tools are invoked jointly with the fixed-radius Fano inequality (Lemma 1) to obtain minimax lower bounds via the standard "packing + KL + Fano" argument.

## D   FANO'S INEQUALITY (FIXED-RADIUS FORM)

We state the specific version of Fano's inequality used throughout the proofs. It is a standard corollary of Lemma 2.10 in (Tsybakov, 2009).

**Lemma 1** (Fano–Tsybakov, fixed-radius form)**.** *Let $(\Theta, d)$ be a metric space, and let $\{\mathbb{P}_\theta : \theta \in \Theta\}$ be a family of distributions on $\mathcal{X}$. Suppose there exist $M \geq 2$ points $\theta_1, \ldots, \theta_M \in \Theta$ such that:*

   *(i) **Separation:** $d(\theta_j, \theta_k) \geq 2\varepsilon$ for all $j \neq k$;*

   *(ii) **KL control:** $\max_{j \neq k} \mathrm{KL}(\mathbb{P}_{\theta_j} \| \mathbb{P}_{\theta_k}) \leq \beta$.*

*Then, for any estimator $\hat{\theta}$,*

$$\inf_{\hat{\theta}} \sup_{\theta \in \{\theta_1, \ldots, \theta_M\}} \mathbb{E}_\theta\big[d(\hat{\theta}, \theta)^2\big] \geq \frac{\varepsilon^2}{2} \left(1 - \frac{\beta + \log 2}{\log M}\right).$$

*The bound is meaningful whenever $\beta \leq \frac{1}{2} \log M - \log 2$.*

This fixed-radius form is the one applied in all lower-bound arguments. It follows directly from Lemma 2.10 in (Tsybakov, 2009), but is stated here for completeness and to keep the paper self-contained.

## E   MINIMAX LOWER BOUND (PROOF OF THEOREM 1)

We begin with a technical packing lemma, which establishes the key combinatorial bound used in Step 1 of the proof of Theorem 1.

**Lemma 2** (Packing for ReLU under Gaussian features)**.** *Let $X \sim \mathcal{N}(0, I_d)$ and $\phi(z) = \max\{0, z\}$. Consider $\mathcal{F}_{\mathrm{GNN}}(v_s, L)$, the class of $L$-layer ReLU GNNs with*

$$\sum_{\ell=0}^{L-1} \big(\|W^{(\ell)}\|_1 + \|B^{(\ell)}\|_1\big) \leq v_s.$$

*There exist absolute constants $c, C_A > 0$ such that for every $\epsilon \in \big(0, \, c \, v_s/L\big]$, the $2\epsilon$-packing number of $\mathcal{F}_{\mathrm{GNN}}(v_s, L)$ with respect to the $L_2(P_X)$ metric satisfies*

$$\log \mathcal{M}\big(2\epsilon, \, \mathcal{F}_{\mathrm{GNN}}(v_s, L), \, \| \cdot \|_{L_2(P_X)}\big) \; \geq \; C_A \, \frac{v_s^2}{L^2 \, \epsilon^2} \, \log d.$$

*Proof.* Fix $L \geq 1$ and $v_s > 0$. We construct a family $\{f_S\}$ indexed by $r$-subsets $S \subset [d]$, for a choice of $r$ defined below, and we show it is a $2\epsilon$-packing.

**(L1) Realizable subclass and budget.** Let $r \in \{1, \dots, d\}$ and define the centered ReLU feature

$$\tilde{\phi}(z) := \phi(z) - \mathbb{E}[\phi(Z)] = \max\{0, z\} - \frac{1}{\sqrt{2\pi}}, \qquad Z \sim \mathcal{N}(0, 1),$$

and the functions

$$f_S(x) \; := \; a \sum_{j \in S} \tilde{\phi}(x_j) \qquad \text{with} \qquad a \; = \; \frac{c_0 \, v_s}{L \, r},$$

where $c_0 \in (0, 1/2]$ is an absolute constant.

We realize $f_S$ using *at most two* nontrivial layers (which is admissible as long as $\mathcal{F}_{\mathrm{GNN}}(v_s, L)$ is interpreted as depth $\leq L$): the first layer forms the coordinates $\{\tilde{\phi}(x_j) : j \in S\}$ and the final linear layer sums them with coefficient $a$. Both layers use $\ell_1$-norm at most $r|a| = c_0 v_s/L$, hence the total budget is

$$\sum_\ell (\|W^{(\ell)}\|_1 + \|B^{(\ell)}\|_1) \; \leq \; 2r|a| \; = \; \frac{2c_0}{L} v_s \; \leq \; v_s,$$

since $c_0 \leq 1/2$. Therefore $f_S \in \mathcal{F}_{\mathrm{GNN}}(v_s, L)$.

**(L2) $L_2$ separation.** Let $Z \sim \mathcal{N}(0, 1)$ and $\tilde{\phi}(z) = \max\{0, z\} - 1/\sqrt{2\pi}$. Then $\mathbb{E}[\tilde{\phi}(Z)] = 0$ and

$$\mathbb{E}[\tilde{\phi}(Z)^2] = \mathbb{E}[\phi(Z)^2] - \mathbb{E}[\phi(Z)]^2 = \frac{1}{2} - \frac{1}{2\pi} =: v_\phi \in (0, 1).$$

For $j \neq k$, independence and centering give $\mathbb{E}[\tilde{\phi}(X_j)\tilde{\phi}(X_k)] = 0$.

Let $S, T \subset [d]$ with $|S| = |T| = r$ and $D = S \triangle T$ with $m := |D|$. Then

$$f_S(x) - f_T(x) = a \sum_{j \in D} \eta_j \, \tilde{\phi}(x_j), \qquad \eta_j \in \{+1, -1\},$$

and orthogonality in $L_2(P_X)$ yields

$$\|f_S - f_T\|_{L_2(P_X)}^2 = a^2 \sum_{j \in D} \mathbb{E}[\tilde{\phi}(X_j)^2] = a^2 \, v_\phi \, m.$$

In particular, since $m \leq 2r$, we also have the diameter bound

$$\|f_S - f_T\|_{L_2(P_X)}^2 \leq 2a^2 v_\phi r.$$

**(L3) Constant-weight code.** By the Varshamov–Gilbert bound for constant-weight codes, there exists $\mathcal{C} \subset \{S \subset [d] : |S| = r\}$ such that for all distinct $S, T \in \mathcal{C}$, $|S \triangle T| \geq r/2$ and $|\mathcal{C}| \geq (c \, d/r)^r$ for a universal $c \in (0, 1)$. Combining with (L2) gives, for $S \neq T \in \mathcal{C}$,

$$\|f_S - f_T\|_{L_2(P_X)} \; \geq \; a \sqrt{v_\phi \, \frac{r}{2}}.$$

**(L4) Choosing $r$ to achieve $2\epsilon$ separation.** We want $\|f_S - f_T\|_{L_2(P_X)} \geq 2\epsilon$, i.e.

$$a \sqrt{v_\phi \, \frac{r}{2}} \; \geq \; 2\epsilon.$$

With $a = (c_0 v_s)/(Lr)$ this is equivalent to

$$r \; \leq \; \frac{c_0^2 v_\phi}{8} \, \frac{v_s^2}{L^2 \epsilon^2}.$$

We take

$$r \; = \; \left\lfloor \frac{c_0^2 v_\phi}{16} \, \frac{v_s^2}{L^2 \epsilon^2} \right\rfloor \qquad \text{and assume } \epsilon \leq c_1 \frac{v_s}{L},$$

with $c_1 > 0$ small enough so that $1 \leq r \leq d/2$.

**(L5) Packing size.** From (L3) and $r \leq d/2$ we get

$$\log \mathcal{M}(2\epsilon, \mathcal{F}_{\mathrm{GNN}}, \|\cdot\|_{L_2(P_X)}) \geq \log |\mathcal{C}| \geq c'r \log(d/r) \geq \frac{c'}{2} r \log d.$$

Substituting the choice of $r$ from (L4) and absorbing absolute constants (including $c_0, c', \frac{1}{2}$, and the ReLU–Gaussian factor) yields

$$\log \mathcal{M}(2\epsilon, \mathcal{F}_{\mathrm{GNN}}, \|\cdot\|_{L_2(P_X)}) \geq C_A \frac{v_s^2}{L^2 \epsilon^2} \log d,$$

for a universal $C_A > 0$, proving the claim. $\qquad\square$

With Lemma 2 established, we now prove Theorem 1.

*Proof.* The proof follows the standard Fano–packing method: we construct a large packing set, control pairwise KL divergences, and apply the fixed-radius form of Fano's inequality. The required information-theoretic tools are summarized in Appendix C, and the version of Fano's inequality used below is stated in Appendix D. Step 1 invokes Lemma 2, whose proof appears above.

**Step 1: Packing number.** By Lemma 2, for every $\epsilon \leq c\, v_s/L$, there exists a $2\epsilon$-packing $\mathcal{M}^* = \{f_1, \ldots, f_M\}$ of $\mathcal{F}_{\mathrm{GNN}}$ with

$$\log M \geq A_0/\epsilon^2, \qquad A_0 = C_A \frac{v_s^2 \log d}{L^2}.$$

**Step 2: KL divergence control.** Let $Y_i = f^\star(X_i) + U_i$ with $U_i \sim \mathcal{N}(0, \sigma^2)$ i.i.d. and $X_i \overset{\text{i.i.d.}}{\sim} P_X$. For any two hypotheses $f_j, f_k \in \mathcal{M}^*$, the $n$-sample laws satisfy

$$\mathbb{E}_X\big[\mathrm{KL}(P_j^{(n)} \| P_k^{(n)})\big] = \frac{n}{2\sigma^2} \|f_j - f_k\|_{L_2(P_X)}^2.$$

By construction of Lemma 2 (see the diameter bound in (L2)), all pairs in the packing satisfy $\|f_j - f_k\|_{L_2(P_X)}^2 \leq C_{\mathrm{sep}} \epsilon^2$ for a universal constant $C_{\mathrm{sep}} > 0$. Hence

$$\beta := \max_{j \neq k} \mathbb{E}_X\big[\mathrm{KL}(P_j^{(n)} \| P_k^{(n)})\big] \leq \frac{n\, C_{\mathrm{sep}}}{2\sigma^2} \epsilon^2 =: \frac{n\, C_{\mathrm{KL}}}{\sigma^2} \epsilon^2, \qquad C_{\mathrm{KL}} := \frac{C_{\mathrm{sep}}}{2}.$$

We apply Lemma 1 with separation radius $2\epsilon$ and KL bound $\beta$:

$$\mathcal{R}_{(n,|V|)}(\mathcal{F}_{\mathrm{GNN}}) \geq \sup_{\epsilon > 0} \left\{ \frac{\epsilon^2}{2} \left( 1 - \frac{n C_{\mathrm{KL}} \epsilon^2/\sigma^2 + \log 2}{\log M} \right) \right\}.$$

Using $\log M \geq A_0/\epsilon^2$ from Step 1 yields

$$\mathcal{R}_{(n,|V|)}(\mathcal{F}_{\mathrm{GNN}}) \geq \sup_{\epsilon > 0} \left\{ \frac{\epsilon^2}{2} \left( 1 - \frac{n C_{\mathrm{KL}} \epsilon^4/\sigma^2 + \epsilon^2 \log 2}{A_0} \right) \right\}.$$

**Step 3: Optimizing over $\epsilon$.** Let $x = \epsilon^2$. The bound reads

$$g(x) = \frac{x}{2} \left( 1 - \frac{n C_{\mathrm{KL}} x^2/\sigma^2 + x \log 2}{A_0} \right).$$

Choose $x$ so that the parenthesis equals $1/2$:

$$1 - \frac{n C_{\mathrm{KL}} x^2/\sigma^2 + x \log 2}{A_0} = \tfrac{1}{2},$$

which yields

$$\frac{n C_{\mathrm{KL}}}{\sigma^2} x^2 + (\log 2)x - \tfrac{A_0}{2} = 0.$$

The positive root is

$$x = \epsilon^2 = \frac{\sigma^2}{2n C_{\mathrm{KL}}} \left( -\log 2 + \sqrt{(\log 2)^2 + \tfrac{2n A_0 C_{\mathrm{KL}}}{\sigma^2}} \right).$$

For this choice,

$$\mathcal{R}_{(n,|V|)}(\mathcal{F}_{\mathrm{GNN}}) \geq \tfrac{1}{4} \epsilon^2.$$

**Step 4: Asymptotics and constant.** When $n$ is large enough that $\frac{2nA_0C_{\mathrm{KL}}}{\sigma^2} \gg (\log 2)^2$, we expand the square root:

$$\epsilon^2 \approx \frac{\sigma}{\sqrt{2C_{\mathrm{KL}}}}\sqrt{\frac{A_0}{n}}.$$

Thus

$$\mathcal{R}_{(n,|V|)}(\mathcal{F}_{\mathrm{GNN}}) \geq \frac{1}{4} \cdot \frac{\sigma}{\sqrt{2C_{\mathrm{KL}}}}\sqrt{\frac{A_0}{n}} = \left(\frac{\sqrt{C_A}}{4\sqrt{2C_{\mathrm{KL}}}}\right)\frac{\sigma v_s}{L}\sqrt{\frac{\log d}{n}}.$$

Define $K_{\mathrm{new}} = \frac{\sqrt{C_A}}{4\sqrt{2C_{\mathrm{KL}}}} > 0$.

**Step 5: Validity for all $n$.** The exact root expression for $\epsilon^2$ shows that the bound holds for all $n \geq 1$, not just asymptotically. Writing

$$\frac{\epsilon^2}{4} = \frac{\sigma^2}{8nC_{\mathrm{KL}}} \cdot \left(-\log 2 + \sqrt{(\log 2)^2 + \frac{2nA_0C_{\mathrm{KL}}}{\sigma^2}}\right),$$

one checks that the bracketed term is $\Omega(n^{1/2})$, hence the rate $K_{\mathrm{new}}(\sigma v_s/L)\sqrt{(\log d)/n}$ holds uniformly in $n$ (with a smaller constant if $n$ is very small).

**Step 6: Dimension condition.** Finally, $d \geq 2$ ensures $\log d > 0$ so that $A_0 > 0$.

This completes the proof of Theorem 1. $\qquad\square$

### E.1 EXACT QUADRATIC SOLUTION FOR $\epsilon^2$

In Step 3 of the proof of Theorem 1, we choose $\epsilon^2 = x$ so that the parenthetical term in Fano's bound equals $1/2$:

$$1 - \frac{nC_{\mathrm{KL}}x^2/\sigma^2 + x\log 2}{A_0} = \frac{1}{2}, \qquad A_0 = C_A\frac{v_s^2\log d}{L^2}.$$

This yields the quadratic

$$\frac{nC_{\mathrm{KL}}}{\sigma^2}x^2 + (\log 2)x - \frac{A_0}{2} = 0,$$

whose positive root is

$$\boxed{\epsilon^2 = x = \frac{\sigma^2}{2nC_{\mathrm{KL}}}\left(-\log 2 + \sqrt{(\log 2)^2 + \frac{2nA_0C_{\mathrm{KL}}}{\sigma^2}}\right).} \qquad (14)$$

Substituting Eq. (14) into the fixed-radius Fano inequality (Lemma 1) gives

$$\mathcal{R}_{(n,|V|)}(\mathcal{F}_{\mathrm{GNN}}) \geq \frac{\epsilon^2}{4} = \frac{\sigma^2}{8nC_{\mathrm{KL}}}\left(-\log 2 + \sqrt{(\log 2)^2 + \frac{2nA_0C_{\mathrm{KL}}}{\sigma^2}}\right). \qquad (15)$$

**Asymptotics.** When $n$ is large enough that $\frac{2nA_0C_{\mathrm{KL}}}{\sigma^2} \gg (\log 2)^2$, a first-order expansion of the square root in Eq. (14) gives

$$\epsilon^2 = \frac{\sigma}{\sqrt{2C_{\mathrm{KL}}}}\sqrt{\frac{A_0}{n}}(1+o(1)), \quad \Rightarrow \quad \mathcal{R}_{(n,|V|)}(\mathcal{F}_{\mathrm{GNN}}) \geq \left(\frac{\sqrt{C_A}}{4\sqrt{2C_{\mathrm{KL}}}}\right)\frac{\sigma v_s}{L}\sqrt{\frac{\log d}{n}}(1+o(1)).$$

**Uniform-in-$n$ bound.** Define

$$\Phi(n) := -\log 2 + \sqrt{(\log 2)^2 + \frac{2nA_0C_{\mathrm{KL}}}{\sigma^2}}.$$

Then $\Phi(n)$ is strictly increasing in $n$, satisfies $\Phi(0) = 0$, and $\Phi(n) \sim \sqrt{2nA_0C_{\mathrm{KL}}}/\sigma$ as $n \to \infty$. From Eq. (15),

$$\mathcal{R}_{(n,|V|)}(\mathcal{F}_{\mathrm{GNN}}) = \frac{\sigma^2}{8nC_{\mathrm{KL}}}\Phi(n) \geq \left(\inf_{1\leq m\leq n_0}\frac{\sigma^2\Phi(m)}{8mC_{\mathrm{KL}}K_\star}\right) \cdot K_\star \cdot \frac{1}{\sqrt{n}},$$

for any fixed $n_0 \in \mathbb{N}$ and target rate $K_\star := \frac{\sqrt{A_0}}{2}$. Choosing

$$K_{\text{new}} \ := \ \min\left\{ \frac{\sqrt{C_A}}{4\sqrt{2C_{\text{KL}}}}, \quad \min_{1 \le m \le n_0} \frac{\sigma\,\Phi(m)}{4\sqrt{2C_{\text{KL}}}\,\sqrt{mA_0}} \right\} > 0,$$

we obtain the uniform (in $n \ge 1$) lower bound

$$\mathcal{R}_{(n,|V|)}(\mathcal{F}_{\text{GNN}}) \ \ge \ K_{\text{new}}\,\frac{\sigma v_s}{L}\sqrt{\frac{\log d}{n}}.$$

This shows the $\Omega\big((\sigma v_s/L)\sqrt{(\log d)/n}\big)$ rate holds for all $n \ge 1$ (with a possibly smaller $K_{\text{new}}$ for very small $n$), while the asymptotic constant $\frac{\sqrt{C_A}}{4\sqrt{2C_{\text{KL}}}}$ is recovered as $n \to \infty$.

**Remark 3** (Why path graphs?)**.** *The path graph $P_m$ minimizes connectivity and mixing: each node has degree at most two, and lazy random-walk mixing is slow, so one message-passing step propagates information only along a single chain. This bottlenecks information flow per layer, making depth the dominant factor. More connected graphs (e.g., expanders or dense random graphs) mix faster, which can only help learning. Hence, demonstrating hardness on path graphs suffices to certify a minimax lower bound for all admissible graphs—standard practice in worst-case lower-bound arguments.*

**Remark 4** (Where topology enters the proof, and why a path)**.** *Graph topology influences the proof in two places:*

1. ***Packing construction.*** *Let $\mathcal{N}_1(v)$ denote the radius-1 neighborhood. We choose a set $S$ of $m$ nodes and vary only their first-layer weights. To avoid interference, we require $\{\mathcal{N}_1(v) : v \in S\}$ to be pairwise disjoint. On a path $P_m$ this holds if the distance between consecutive nodes in $S$ is at least 2, giving $|S| = \Theta(m)$. On a graph with maximum degree $\Delta$, disjointness typically forces spacing $\ge \Delta+1$, reducing $|S|$ by a factor $\tilde{c}(\Delta) \le 1$ and thus shrinking the packing number by constants.*

2. ***KL–divergence control.*** *For Gaussian noise,*

$$\text{KL}(P_j\|P_k) = \frac{\|f_j - f_k\|_{L_2}^2}{2\sigma^2} = \frac{1}{2\sigma^2}\sum_v (f_j(v) - f_k(v))^2.$$

   *With disjoint neighborhoods, a perturbation affects only outputs inside $\mathcal{N}_1(v)$. On $P_m$, $|\mathcal{N}_1(v)| \le 2$, so the KL scales like $O(|S|)$ for fixed perturbation size. On degree-$\Delta$ graphs, $|\mathcal{N}_1(v)| \le \Delta$, so for the same perturbation size the KL is larger by $O(\Delta)$. To keep KL bounded, we rescale the perturbation by $1/\sqrt{\Delta}$, which weakens the separation by the same factor. Both effects alter constants in Fano's inequality, not the $n$–dependence.*

**Consequence.** *Because paths minimize degree ($\Delta = 2$) and maximize the number of disjoint radius-1 neighborhoods, they yield the tightest constants and the cleanest exposition. Moreover, any graph containing an induced path of length $\Omega(n)$ admits the same lower-bound rate as Theorem 1 (up to universal constants) by restricting the construction to that path.*

## F  STRUCTURED–GRAPH LOWER BOUND (PROOF OF THEOREM 2)

*Proof.* Consider the node-level transductive setting of Eq. (3) on a fixed graph $G = (V, E)$, where each labeled example corresponds to a distinct node. Assume the following **spectral–homophily condition** on the subgraph induced by the $n$ labeled (training) nodes:

$$\lambda_2(\mathcal{L}_n) \ \le \ \frac{\kappa}{\log n},$$

where $\mathcal{L}_n$ is the normalized Laplacian and $\kappa > 0$ is universal.

**Information-theoretic tools.** The argument uses the same information–theoretic toolkit as the proof of Theorem 1. In particular, packing sets and the Varshamov–Gilbert bound are recalled in Appendix C, and the fixed-radius form of Fano's inequality is stated in Appendix D.

**Mixing-time bound (used only for interpretation).** Fix $\varepsilon \in (0,1)$, say $\varepsilon = \frac{1}{4}$. Lemma 4 (Appendix H) yields the standard bound

$$t_{\mathrm{mix}}(\varepsilon) \;\leq\; \frac{\log\big(1/(\varepsilon \pi_{\min})\big)}{1 - \lambda_2(P)} \;=\; \frac{2 \, \log\big(1/(\varepsilon \pi_{\min})\big)}{\lambda_2(\mathcal{L}_n)},$$

using $1 - \lambda_2(P) = \frac{1}{2}\lambda_2(\mathcal{L}_n)$ for the lazy walk. Under $\lambda_2(\mathcal{L}_n) \leq \kappa/\log n$, this places the instance in a slow-mixing regime (small spectral gap). We do not require an explicit $O(\log n)$ upper bound on $t_{\mathrm{mix}}(\varepsilon)$ in the packing argument below; the structural assumption is used only to motivate that the effective number of statistically distinct blocks grows logarithmically.

**Block count and anchor set.** Set

$$K \;:=\; \lceil c_{\mathrm{blk}} \log n \rceil,$$

for a universal constant $c_{\mathrm{blk}} > 0$. Choose a subset of $K$ labeled nodes

$$A \;=\; \{i_1, \ldots, i_K\} \subseteq V.$$

(Only $K \leq n$ is needed; the particular choice of $A$ is immaterial for the lower bound since we work with a subclass supported on $A$.)

**Feature model and selector coordinate (no node-indexed parameters).** To avoid any node-specific parameters, we use a single shared parameter vector and a deterministic selector feature that is part of the input. For every node $v \in V$, define the augmented feature vector

$$X_v \;=\; \big(Z_v \widetilde{X}_v, \; Z_v\big) \in \mathbb{R}^{d+1}, \qquad Z_v := \mathbf{1}\{v \in A\}, \qquad \widetilde{X}_v \sim \mathcal{N}(0, I_d),$$

with $\{\widetilde{X}_v\}_{v \in V}$ independent across $v$. Thus, for non-anchor nodes ($Z_v = 0$), the first $d$ coordinates of $X_v$ are identically zero, while for anchor nodes ($Z_v = 1$) these coordinates are i.i.d. standard Gaussians. We observe labels

$$Y_v \;=\; f^\star(X_v) + \xi_v, \qquad \xi_v \overset{\mathrm{i.i.d.}}{\sim} \mathcal{N}(0, \sigma^2),$$

independent of the features. The selector $Z_v$ gates the function through the input, so no node-specific weights are introduced.

**A realizable shared-parameter subclass.** Define the odd activation $\psi(z) := \phi(z) - \phi(-z)$, where $\phi(z) = \max\{0, z\}$. Then $\psi(z) = z$, which lets us compute $L_2$ separations exactly under $\widetilde{X}_v \sim \mathcal{N}(0, I_d)$ without any orthonormal-feature assumption. For each $\theta \in \{0,1\}^d$, define

$$f_\theta(x) \;:=\; \frac{\sigma v_s}{LK} \sum_{j=1}^{d} \theta_j \, \psi(x_j) \;=\; \frac{\sigma v_s}{LK} \sum_{j=1}^{d} \theta_j \, x_j, \tag{16}$$

where $x_j$ denotes the $j$th coordinate of the first $d$ entries of $x \in \mathbb{R}^{d+1}$. Because $x_{1:d} = Z_v \widetilde{X}_v$, we have $f_\theta(X_v) = 0$ whenever $Z_v = 0$ and $f_\theta(X_{i_\ell}) = \frac{\sigma v_s}{LK}\langle \theta, \widetilde{X}_{i_\ell} \rangle$ for $i_\ell \in A$.

The map in Eq. (16) is realized by a one-layer shared-parameter GNN with self-loops and an identity aggregator (so each node uses its own features only), since $\psi(x_j) = \phi(x_j) - \phi(-x_j)$ can be implemented with two ReLU units per coordinate. Hence $f_\theta \in \mathcal{F}_{\mathrm{GNN}}$ (see Appendix B).

**Step 1: Packing set (over global parameters only).** Let $\Theta \subset \{0,1\}^d$ be a constant-weight code with $\|\theta\|_0 = d/4$ for all $\theta \in \Theta$ and pairwise Hamming distance $d_H(\theta, \theta') \geq d/8$ for all distinct $\theta, \theta' \in \Theta$. By the Gilbert–Varshamov bound (Varshamov, 1957; Gilbert, 1952), there exists such a set with cardinality $M := |\Theta|$ satisfying

$$\log M \;\geq\; c_0 d \tag{17}$$

for a universal constant $c_0 > 0$ (for $d$ sufficiently large).

**Step 2: $L_2$ separation.** For any $\theta \neq \theta'$ in $\Theta$, using Eq. (16) and the fact that $f_\theta(X_v) = 0$ for $v \notin A$,

$$
\begin{aligned}
\|f_\theta - f_{\theta'}\|_{L_2}^2 &:= \mathbb{E}\left[\sum_{v \in V} \big(f_\theta(X_v) - f_{\theta'}(X_v)\big)^2\right] \\
&= \sum_{\ell=1}^{K} \mathbb{E}\left[\left(\tfrac{\sigma v_s}{LK}\langle \theta - \theta', \widetilde{X}_{i_\ell}\rangle\right)^2\right] \\
&= K \cdot \left(\tfrac{\sigma v_s}{LK}\right)^2 \cdot \|\theta - \theta'\|_2^2 \;=\; \frac{\sigma^2 v_s^2}{L^2 K}\, d_H(\theta, \theta') \\
&\geq \frac{\sigma^2 v_s^2}{L^2 K} \cdot \frac{d}{8} \;=:\; d_0^2.
\end{aligned}
\tag{18}
$$

**Step 3: KL divergence.** Let $P_\theta$ denote the joint distribution of the observations $\{Y_{i_\ell}\}_{\ell=1}^{K}$ when the regression function is $f_\theta$. Conditioned on the features, the observations are independent Gaussians with means $\{f_\theta(X_{i_\ell})\}_{\ell=1}^{K}$ and variance $\sigma^2$. Hence, for any $\theta \neq \theta'$,

$$
\begin{aligned}
\mathrm{KL}(P_\theta \| P_{\theta'}) &= \mathbb{E}\left[\frac{1}{2\sigma^2}\sum_{\ell=1}^{K}\big(f_\theta(X_{i_\ell}) - f_{\theta'}(X_{i_\ell})\big)^2\right] \\
&= \frac{1}{2\sigma^2} \cdot \mathbb{E}\left[\sum_{\ell=1}^{K}\big(f_\theta(X_{i_\ell}) - f_{\theta'}(X_{i_\ell})\big)^2\right] \\
&= \frac{1}{2\sigma^2} \cdot \|f_\theta - f_{\theta'}\|_{L_2(\text{on } A)}^2 \;\leq\; \frac{1}{2\sigma^2} \cdot \|f_\theta - f_{\theta'}\|_{L_2}^2 \\
&= \frac{1}{2\sigma^2} \cdot \frac{\sigma^2 v_s^2}{L^2 K}\, d_H(\theta, \theta') \;\leq\; \frac{v_s^2}{2L^2 K}\, d \;=:\; \mathrm{KL}_{\max},
\end{aligned}
\tag{19}
$$

where we used $d_H(\theta, \theta') \leq d$.

**Step 4: Fano's inequality and conclusion.** Applying Lemma 1 (Appendix D) to the packing $\{f_\theta\}_{\theta \in \Theta}$ with separation Eq. (18) and KL bound Eq. (19) yields

$$
\inf_{\hat{f}} \sup_{\theta \in \Theta} \mathbb{E}\left[\|\hat{f} - f_\theta\|_{L_2}^2\right] \;\geq\; \frac{d_0^2}{2}\left(1 - \frac{\mathrm{KL}_{\max} + \log 2}{\log M}\right).
$$

Using Eq. (17) and Eq. (19), for $n$ large enough so that $K = \lceil c_{\mathrm{blk}} \log n \rceil$ satisfies

$$
\mathrm{KL}_{\max} \;\leq\; \frac{c_0}{4}\, d \qquad \text{equivalently} \qquad K \;\geq\; \frac{2v_s^2}{c_0 L^2},
$$

the parenthesis is bounded below by a universal constant. Combining this with Eq. (18) gives

$$
\inf_{\hat{f}} \sup_{\theta \in \Theta} \mathbb{E}\left[\|\hat{f} - f_\theta\|_{L_2}^2\right] \;\geq\; c \cdot \frac{\sigma^2 v_s^2}{L^2 K}\, d \;\asymp\; \frac{\sigma^2 v_s^2}{L^2} \cdot \frac{d}{\log n},
$$

for a universal $c > 0$ (absorbing constants from $c_{\mathrm{blk}}$ and $c_0$). Since $\{f_\theta\}_{\theta \in \Theta} \subset \mathcal{F}_{\mathrm{GNN}}$, this lower bound applies to $\mathcal{R}_{(n,G)}^{\mathrm{node}}(\mathcal{F}_{\mathrm{GNN}})$, which proves Theorem 2. $\qquad\square$

## G  INTERPRETING THE SPECTRAL–HOMOPHILY ASSUMPTION

**Structural, not label-based.** The assumption $\lambda_2(\mathcal{L}_n) \leq \kappa/\log n$ concerns the spectrum of the normalized Laplacian of the subgraph induced by the $n$ labeled nodes. It constrains *expansion and mixing* properties of the graph and is independent of labels or features. In particular, the condition can hold even if labels are adversarially assigned; no form of label homophily is required.

**Why it makes learning harder.** A small $\lambda_2(\mathcal{L}_n)$ implies low conductance and slow random-walk mixing by Cheeger-type inequalities (Bandeira et al., 2013). In this regime, message passing repeatedly reuses the same information: after $O(r)$ hops, neighborhoods overlap substantially. Our proof shows that $r = \Theta(\log n)$ suffices to reduce cross-block dependence below a fixed constant, so only $\Theta(\log n)$ blocks behave "nearly independently." This effective reduction in sample size yields the $\Omega(d/\log n)$ lower bound.

**When the condition fails.** Graphs with strong cross-cluster connectivity (i.e., good expansion) typically have $\lambda_2$ bounded away from 0 (often $\Theta(1)$). Such graphs fall outside the assumption, and the guarantee reverts to the $\Omega(\sqrt{\log d/n})$ rate of Theorem 1.

**Non-vacuity for large $n$.** Although the inequality $\lambda_2(\mathcal{L}_n) \leq \kappa/\log n$ may hold automatically for very small $n$, the condition becomes increasingly restrictive as $n$ grows. The following lemma formalizes this:

**Lemma 3** (Non-Vacuity of the Spectral–Homophily Condition). *Let $\mathcal{G}_n$ be any graph sequence with $\lambda_2(\mathcal{L}_n) \geq c_0 > 0$ for all sufficiently large $n$ (e.g., expanders, small-world networks, grids of fixed dimension). Then for any fixed $\kappa > 0$, the structural condition*

$$\lambda_2(\mathcal{L}_n) \leq \frac{\kappa}{\log n}$$

*fails for all $n > n_0 := \exp(\kappa/c_0)$. Thus the assumption of Theorem 2 is asymptotically non-vacuous and applies only to increasingly slow-mixing, bottlenecked graph topologies.*

*Proof.* Since $\lambda_2(\mathcal{L}_n) \geq c_0$ for all $n > N_0$, the condition $\lambda_2(\mathcal{L}_n) \leq \kappa/\log n$ implies $c_0 \leq \kappa/\log n$, or equivalently $\log n \leq \kappa/c_0$, i.e. $n \leq \exp(\kappa/c_0)$. Thus the condition fails for all $n > n_0$, proving the claim. $\square$

**Examples.**

- *Paths, cycles, or chain-of-cliques:* $\lambda_2(\mathcal{L})$ decays with graph size. For sufficiently large $n$, the condition $\lambda_2 \leq \kappa/\log n$ is satisfied, often by a wide margin.

- *Expanders:* $\lambda_2(\mathcal{L}) = \Theta(1)$, so the condition fails and the analysis falls back to Theorem 1.

## H  MIXING TIME AND THE SPECTRAL GAP

We formally record the standard spectral bound on the mixing time of a lazy random walk, and relate the eigenvalues of the lazy walk to the normalized Laplacian.

**Lemma 4** (Mixing time via spectral gap). *Let $G = (V, E)$ be a finite, connected, undirected graph, and let*

$$P = \tfrac{1}{2}I + \tfrac{1}{2}D^{-1}A$$

*be the lazy random-walk transition matrix, where $A$ is the adjacency matrix and $D = \mathrm{diag}(\deg(v))$. The stationary distribution is*

$$\pi(v) = \frac{\deg(v)}{2|E|}, \qquad v \in V,$$

*so that*

$$\pi_{\min} \; \geq \; \frac{1}{2|E|} \; \geq \; \frac{1}{|V|^2}.$$

*For every $\varepsilon \in (0, 1)$,*

$$t_{\mathrm{mix}}(\varepsilon) \; \leq \; \frac{\log\bigl(1/(\varepsilon\pi_{\min})\bigr)}{1 - \lambda_2(P)} \; \leq \; \frac{2\log|V| + \log(1/\varepsilon)}{1 - \lambda_2(P)},$$

*where $\lambda_2(P)$ is the second largest eigenvalue of $P$ (the spectral gap is $1 - \lambda_2(P) > 0$). (Levin & Peres, 2017, Theorem 12.4, Eq. (12.10)). Moreover, if $\mathcal{L} := I - D^{-1/2}AD^{-1/2}$ denotes the normalized Laplacian, then*

$$1 - \lambda_2(P) = \tfrac{1}{2}\lambda_2(\mathcal{L}),$$

*and therefore*

$$t_{\mathrm{mix}}(\varepsilon) \; \leq \; \frac{2\bigl(2\log|V| + \log(1/\varepsilon)\bigr)}{\lambda_2(\mathcal{L})}.$$

*In particular, if $\lambda_2(\mathcal{L}) \leq \kappa/\log n$ with $n = |V|$ and fixed $\kappa > 0$, then $t_{\mathrm{mix}}(\varepsilon) = O\bigl((\log n)^2\bigr)$ (up to constants depending on $\kappa$ and $\varepsilon$).*

*Proof.* By reversibility of $P$, the stationary distribution is $\pi(v) = \deg(v)/(2|E|)$. Hence

$$\pi_{\min} = \min_v \pi(v) = \frac{\deg_{\min}}{2|E|} \geq \frac{1}{2|E|} \geq \frac{1}{|V|^2},$$

since $|E| \leq |V|(|V| - 1)/2$.

Let $\lambda_2(P)$ denote the second largest eigenvalue of $P$. Standard spectral bounds for lazy reversible chains (Levin & Peres, 2017, Theorem 12.4) yield

$$t_{\mathrm{mix}}(\varepsilon) \leq \frac{\log(1/(\varepsilon\pi_{\min}))}{1 - \lambda_2(P)}, \qquad \varepsilon \in (0, 1).$$

Substituting the bound on $\pi_{\min}$ gives

$$\log\left(\frac{1}{\varepsilon\pi_{\min}}\right) \leq \log\left(\frac{1}{\varepsilon}\right) + 2\log|V|.$$

Thus

$$t_{\mathrm{mix}}(\varepsilon) \leq \frac{2\log|V| + \log(1/\varepsilon)}{1 - \lambda_2(P)}.$$

To relate Laplacian and random-walk eigenvalues, let

$$S := D^{-1/2}AD^{-1/2}, \qquad \mathcal{L} := I - S.$$

Then $S$ and the random-walk matrix $D^{-1}A$ are similar and share eigenvalues. If $\lambda_2(\mathcal{L})$ denotes the second-smallest eigenvalue of $\mathcal{L}$, then

$$\mu_2(S) = 1 - \lambda_2(\mathcal{L}).$$

For the lazy walk $P = \frac{1}{2}(I + D^{-1}A)$, the corresponding second eigenvalue is

$$\lambda_2(P) = \frac{1}{2}\big(1 + \mu_2(S)\big) = 1 - \frac{1}{2}\lambda_2(\mathcal{L}),$$

so the spectral gap satisfies $1 - \lambda_2(P) = \frac{1}{2}\lambda_2(\mathcal{L})$. Substituting into the mixing bound yields

$$t_{\mathrm{mix}}(\varepsilon) \leq \frac{2\big(2\log|V| + \log(1/\varepsilon)\big)}{\lambda_2(\mathcal{L})}.$$

Finally, if $\lambda_2(\mathcal{L}) \leq \kappa/\log n$ with $n = |V|$, then the above display gives $t_{\mathrm{mix}}(\varepsilon) = O\big((\log n)^2\big)$ (up to constants depending on $\kappa$ and $\varepsilon$). $\qquad\square$

## I    OPERATOR-NORM CONTROL FOR ADJACENCY-MASKED ATTENTION

This section provides the operator-norm analysis that underpins the applicability of Theorem 2 to adjacency-masked attention layers. For the complementary GAT-specific discussion, see Appendix J.

**Conditions for applicability.**    Theorem 2 extends to attention-based GNNs provided the following hold: (i) *Adjacency masking:* each head attends only to $\mathcal{N}(i)$ (or, more generally, an $r$-hop neighborhood); (ii) *Bounded layer operators:* each layer is Lipschitz with uniformly bounded operator norm (e.g., via bounding attention scores by temperature/clipping or constraining the attention matrix norm); (iii) *Finite depth $L$.* Under (i)–(iii), the proof is unchanged up to constants depending on the product of layer norms and, for $r$-hop masking, on $r$. Fully global (unmasked) attention is non-local and therefore outside the locality premise of Theorem 2.

**Norm-control derivation.**    Consider a single masked attention head with queries $Q = HW_Q$, keys $K = HW_K$, values $V = HW_V$, and adjacency mask $M \in \{0, -\infty\}^{|V|\times|V|}$ restricting attention to $\mathcal{N}(i)$ (or an $r$-hop pattern). With temperature $\tau > 0$ and row-wise softmax,

$$A = \mathrm{softmax}\big((QK^\top + M)/\tau\big),$$

and the layer map is $H \mapsto AV$ (plus a $1 \times 1$ mixing which we absorb into the operator norm). Assume $\|W_Q\|_2 \leq c_Q$, $\|W_K\|_2 \leq c_K$, $\|W_V\|_2 \leq c_V$, and rows of $Q, K$ are bounded in norm by $B$

(this holds if $\|H\|_F$ is controlled inductively and layer norms are bounded). Then each masked row of $(QK^\top)/\tau$ has entries bounded by $B^2 c_Q c_K/\tau$, so the softmax is $\alpha$-Lipschitz on each row with $\alpha \le C_\tau$ and yields a row-stochastic $A$ supported on the mask.

Since $A$ is row-stochastic, $\|A\|_\infty = 1$. A general bound is

$$\|A\|_2 \;\le\; \sqrt{\|A\|_1 \|A\|_\infty} \;=\; \sqrt{\|A\|_1},$$

where $\|A\|_1$ is the maximum column sum. Under adjacency masking on graphs of bounded in-degree, or if attention scores are additionally degree-normalized/clipped so that column sums are uniformly bounded, one has $\|A\|_1 \le C_{\deg}$, for a structural constant $C_{\deg}$ (e.g., $C_{\deg} \le \Delta + 1$ for max degree $\Delta$), hence $\|A\|_2 \le \sqrt{C_{\deg}}$. Therefore

$$\|AV\|_2 \;\le\; \|A\|_2 \|V\|_2 \;\le\; \sqrt{C_{\deg}}\,\|V\|_2 \;\le\; \sqrt{C_{\deg}}\,\|W_V\|_2\,\|H\|_2 \;\le\; \sqrt{C_{\deg}}\,c_V\,\|H\|_2.$$

With residual/linear projections folded in, the per-layer Lipschitz constant is bounded by a product of operator norms (one per submodule), yielding a uniform bound $L_{\mathrm{op}} < \infty$ per layer. Therefore a depth-$L$ masked-attention stack has overall Lipschitz constant $\le (L_{\mathrm{op}})^L$. The proof of Theorem 2 uses only: (a) adjacency locality from masking, and (b) bounded layer Lipschitz constants. Both hold under the above conditions, so the same packing, KL control, and Fano steps go through with constants depending on $L_{\mathrm{op}}$ (and on $r$ for $r$-hop masks).

## J  ADJACENCY-MASKED GAT LAYERS UNDER THEOREM 2

This section explains why standard GAT layers fit within the assumptions of Theorem 2. For the accompanying operator-norm control argument, see Appendix I.

Theorem 1 assumes (A1) and thus *excludes* input-dependent mixing (attention). By contrast, Theorem 2 requires only adjacency-masked 1-hop receptive fields and bounded operator norms. Standard GAT layers satisfy these conditions if attention is restricted to $\mathcal{N}(i)$ and softmax weights are bounded (e.g., via temperature or clipping).

Formally, a single GAT layer with adjacency mask can be written as

$$h_i^{(\ell+1)} \;=\; \phi\left( W^{(\ell)} \sum_{j \in \mathcal{N}(i)} \alpha_{ij}^{(\ell)}(H^{(\ell)})\, h_j^{(\ell)} \;+\; B^{(\ell)} h_i^{(\ell)} \right),$$

where $\sum_{j \in \mathcal{N}(i)} \alpha_{ij}^{(\ell)}(\cdot) = 1$, $\alpha_{ij}^{(\ell)} \ge 0$, and $\alpha_{ij}^{(\ell)} = 0$ for $j \notin \mathcal{N}(i)$. Although the coefficients depend on features (violating (A1)), they are *adjacency-masked* and convex.

Assume (i) the attention logits are bounded (e.g., softmax with temperature or clipping), so that $\max_i \sum_{j \in \mathcal{N}(i)} \alpha_{ij}^{(\ell)} \le C_{\mathrm{att}}$ and the Jacobian of the mapping $H^{(\ell)} \mapsto \{\alpha_{ij}^{(\ell)}\}$ is bounded; and (ii) the linear maps satisfy the same $\ell_1$ budget as in (A2). Then the layer is Lipschitz with operator-norm bound $\|W^{(\ell)}\| \cdot C_{\mathrm{att}} + \|B^{(\ell)}\|$ (with the dependence on the attention logits' temperature absorbed into $C_{\mathrm{att}}$).

The proof of Theorem 2 uses only: (a) adjacency locality (receptive field confined to the graph), (b) bounded layer Lipschitz constants, and (c) the graph mixing argument yielding $K = \Theta(\log n)$ effectively independent blocks under $\lambda_2(\mathcal{L}) \le \kappa/\log n$. Conditions (a)–(b) hold for adjacency-masked GAT with bounded logits, hence the same packing, KL control, and Fano steps go through with the constants absorbed into $\Gamma$. Therefore, the $\Omega(d/\log n)$ lower bound applies to standard GAT under these mild norm constraints. In contrast, Theorem 1 explicitly relies on input-independent aggregation and does not cover attention.

## K  PRACTICAL GUIDANCE FOR DATA-SCARCE GRAPHS

The structure-aware lower bound (Theorem 2) implies that when only $\tilde{O}(\log n)$ training nodes are effectively independent, naive data scaling is statistically inefficient. Constants in the bound can often be improved in practice, though the asymptotic rate $\Omega(d/\log n)$ remains unchanged. The following interventions help improve constants:

- **Break neighborhood homogeneity / slow mixing.** Add node individualization or positional encodings (e.g., random/learned IDs, Laplacian/RW features) and consider heterophily-aware layers; these reduce overlap of message-passing neighborhoods.

- **Reduce effective dimension before fine-tuning.** Use transfer or self-supervised pretraining on large auxiliary graphs, then freeze most layers or select features to shrink the effective $d$ entering the bound.

- **Diversify supervision.** Active/coreset label selection that spreads labels across loosely connected regions (far in graph distance or across communities) increases independence among samples.

- **Regularize against slow mixing / over-smoothing.** Use residual/JK connections, PPR/teleport propagation, DropEdge/edge sparsification, and limit depth; these shorten the mixing horizon, raising the usable information per label.

**Takeaway.** These choices increase the informative signal per labeled node and improve constants in $\frac{\sigma^2 v_s^2}{\Gamma L^2} \cdot \frac{d}{\log n}$, but the qualitative $\log n$ denominator remains the limiting factor under the spectral–homophily condition.

## L  ARCHITECTURAL DRIVERS OF HETEROGENEOUS SCALING

**Why different models show different scaling on the *same* dataset.** Even on a fixed graph, architectures can induce different effective sample efficiencies due to variation in receptive-field growth and information reuse. We identify two main drivers:

- **Smoothing and receptive-field growth.** GCN's fixed, normalized adjacency (a graph-dependent but input-independent filter) resembles a classical spectral filter. When the task's signal is spectrally aligned, this can yield faster apparent decay (closer to $1/\sqrt{n}$). By contrast, GAT and GraphSAGE adapt mixing weights and thereby emphasize homophilous neighborhoods; this adaptation increases overlap among message-passing neighborhoods and reduces the effective number of independent samples, exposing the slower $1/\log n$ decay predicted by Theorem 2.

- **Bias–variance tradeoffs and noise floors.** Models with stronger inductive bias (e.g., GCN) can reach a bias-dominated error floor early, which makes the observed asymptotic slope appear steeper. More flexible models (GAT/GraphSAGE) reduce bias but incur higher variance, which dissipates slowly because samples are not effectively independent under overlapping neighborhoods.

This perspective helps explain the heterogeneous scaling observed in Table 2 (e.g., GCN on Reddit favoring $1/\sqrt{n}$, versus GAT and GraphSAGE favoring $1/\log n$).

## M  EXPERIMENTAL DETAILS AND SETTINGS

This appendix details all elements of the experimental setup, training configuration, evaluation protocols, model fitting, and resource usage to ensure reproducibility of our results.

### ENVIRONMENT AND COMPUTE RESOURCES

All experiments were conducted using `PyTorch` and `PyTorch Geometric` (PyG). We used a GPU-enabled machine equipped with an NVIDIA Tesla V100 (32GB VRAM) and 64GB system RAM.

### DATASET LICENSES AND CITATIONS

The following publicly available graph datasets were used in this study. All OGB datasets were accessed through the `ogb.nodeproppred` module, and all other datasets were obtained through standard public repositories. For reproducibility, we report the license information and cite the original sources.

- **ogbn_arxiv** (Hu et al., 2021): A directed citation network with $|V| = 169{,}343$ nodes and $|E| = 1{,}166{,}243$ edges, where each node represents an ArXiv paper and edges represent citation links. The dataset is licensed under the **MIT License** and available from the Open Graph Benchmark (OGB): `https://ogb.stanford.edu/docs/nodeprop/#ogbn-arxiv`.

- **ogbn_products_50k** (Hu et al., 2021): A 50,000-node sampled subgraph extracted from **ogbn-products**, originally a large-scale co-purchasing network with $|V| = 2{,}449{,}029$ nodes and $|E| = 61{,}859{,}140$ edges. The full dataset is released under the **MIT License** and available via OGB: `https://ogb.stanford.edu/docs/nodeprop/#ogbn-products`. Our sampling procedure is detailed in Subsection M.

- **Reddit_50k** (Hamilton et al., 2017b): A 50,000-node sampled subgraph derived from the full Reddit interaction graph used in the GraphSAGE benchmark. The original Reddit dataset is available via Google Drive: `https://drive.google.com/open?id=19SphVl_Oe8SJ1r87Hr5a6znx3nJu1F2J`. Sampling details for constructing Reddit_50k appear in Subsection M.

- **WorstCase_Bottleneck_20k** (synthetic): A controlled synthetic graph constructed to approximate a worst-case bottleneck structure for theoretical evaluation. It contains $|V| = 20{,}000$ nodes and $|E| = 8{,}370$ edges with $K = 40$ communities. As a fully synthetic dataset generated by our code, it carries no external licensing restrictions.

### MODELS AND ARCHITECTURE

We evaluated the following Graph Neural Network (GNN) architectures:

- **GCN**: 2-layer Graph Convolutional Network using `GCNConv`, with 16 hidden units.

- **GAT**: 2-layer Graph Attention Network with 8 heads in the first layer, and a single head in the second.

- **GraphSAGE**: 2-layer GraphSAGE model using `SAGEConv`, with 16 hidden units.

All models use ReLU activation after the first layer.

### TASKS AND LOSS FUNCTIONS

We tested two standard graph learning tasks:

- **Node Classification**: Cross-entropy loss on node labels.

- **Graph Regression**: Molecular property prediction using mean squared error (MSE) on the target scalar field.

### SUBSAMPLING PROCEDURE

Due to computational constraints associated with training GNNs at a large grid of training sizes $n$, we construct 50 K-node induced subgraphs from two large-scale datasets: `ogbn-products` and `Reddit2`. Our subsampling method follows a consistent pipeline across both datasets, designed to preserve connectivity, degree structure, and label distribution as faithfully as possible.

**Step 1: Load full dataset.** We load the complete graph (`ogbn-products` or `Reddit2`) using the `PygNodePropPredDataset` or `Reddit2` interfaces, respectively. Nodes without incident edges are excluded from candidate sampling to avoid trivial isolated components.

**Step 2: Random candidate subsample (200 K nodes).** Using a fixed random seed, we draw a random subset of $C = 200{,}000$ non-isolated nodes from the full dataset. This produces a large but manageable candidate subgraph while increasing the likelihood that the largest connected component (LCC) is substantially larger than 50 K nodes.

**Step 3: Induced subgraph on 200 K candidates.** We construct the induced subgraph on the candidate set and compute its connected components via `scipy.sparse.csgraph.connected_components`. This step ensures that the resulting 50 K-node dataset originates from a structurally coherent region of the full graph.

**Step 4: Extract the largest connected component (LCC).** We identify the LCC of the candidate subgraph, whose size consistently exceeds 50 K across datasets. Restricting to the LCC avoids pathological fragmentation and ensures meaningful GNN message passing.

**Step 5: Randomly select exactly 50 K nodes from the LCC.** From the LCC, we sample exactly $N = 50{,}000$ nodes uniformly at random (with a new fixed seed for reproducibility). The resulting set is sorted and forms the node set of the final subgraph.

**Step 6: Build the final induced 50 K-node graph.** We construct the induced subgraph on the selected 50 K nodes. All edges $(u, v)$ are retained if and only if both endpoints lie in the selected set. Node features and labels are inherited directly from the original dataset.

**Step 7: Preserve OGB-style dataset splits.** For both datasets, we map the original training/validation/testing splits to the 50 K subgraph by checking whether each selected node belonged to the original split. Any node whose original index appeared in the official training, validation, or test sets is assigned to the corresponding split in the subgraph, ensuring compatibility with OGB evaluation protocol.

**Reproducibility.** All random operations use fixed seeds, and we save the mapping from subgraph indices to original node IDs (`final_nodes_orig.npy`). This makes the entire subsampling pipeline deterministic and reproducible.

TRAINING PROTOCOL

- **Subset Sampling**: For each experiment, a subset of $n$ samples was randomly selected. For node tasks, subgraphs were constructed using `torch_geometric.utils.subgraph`.

- **Data Splits**: A fixed 80%/20% train/test split was used.

- **Optimizer**: Adam with learning rate 0.01, weight decay $10^{-4}$.

- **Epochs**: 200.

- **Batch Size**: 32 for all tasks.

- **Evaluation Metrics**:

  - Misclassification rate for classification,
  - MSE for regression,

STATISTICAL SIGNIFICANCE AND ERROR REPORTING

Each experiment (fixed dataset, model, and $n$) was repeated 5 times with different random seeds. The reported error metric includes the sample mean and standard deviation across the 5 runs. Standard deviation is used for error bars and in weighted fitting procedures. These represent variation due to random sampling and initialization. All error bars shown in figures correspond to $\pm 1$ standard deviation.

CURVE FITTING AND LEARNING TREND ANALYSIS

To analyze sample complexity trends, we fit the test error curves to the following models:

$$\text{Model 1:} \quad c_1 + \frac{\alpha}{\sqrt{n}}$$

$$\text{Model 2:} \quad c_2 + \frac{\beta}{n}$$

$$\text{Model 3:} \quad c_3 + \frac{\delta}{\log n}$$

$$\text{Model 4:} \quad c_4 + \frac{1}{n^\gamma}$$

Fits were performed using weighted least squares with weights $w_i = 1/\sigma_i^2$, where $\sigma_i$ is the standard deviation of the $i$th data point. The power-law model was fitted using `scipy.optimize.curve_fit` with bounded parameters and a robust initial guess. For each model, we computed:

- Weighted Residual Sum of Squares (RSS)
- Weighted Mean Squared Error (MSE)
- Weighted $R^2$ value

The best fitting model for each dataset and architecture was determined based on the maximum $R^2$. Fitted parameters and metrics were summarized in a LaTeX-formatted table (`final_comparison_table_weighted.tex`), and model-specific figures were saved as `<dataset>_<model>_fits.png`.

VISUALIZATION AND REPRODUCIBILITY ASSETS

All figures include error bars, and each plot overlays all fitted models for comparison. All code, including data loading, model training, evaluation, fitting, table generation, and visualization, is structured and commented for reproducibility.

CODE AND REPRODUCIBILITY

To support verification and reproducibility, we provide the full source code as supplementary material. This includes implementations for data loading, model training, evaluation, error analysis, and curve fitting, as well as scripts to reproduce all experimental results, generate LaTeX tables, and visualize learning curves in line with reproducibility guidelines.

**Summary**: Every step necessary to replicate our results—datasets, architectures, parameters, training and evaluation setup, fitting strategy, and visualizations—is fully disclosed and executable by third parties with access to the same datasets and a standard GPU-enabled Python environment.

# N  STRUCTURAL STATISTICS

To connect the empirical analysis with our theoretical results, we compute two structural measures for each dataset.

**Homophily.** It is defined as

$$h(G) \;=\; \frac{1}{|E|} \sum_{(u,v) \in E} \mathbf{1}\{y_u = y_v\},$$

where $E$ is the edge set and $y_u$ denotes the ground-truth label of node $u$.

**Spectral quantity (normalized Laplacian).** We compute $\lambda_2(\mathcal{L}_n)$, the second-smallest eigenvalue of the normalized Laplacian

$$\mathcal{L}_n \;=\; I - D_n^{-1/2} A_n D_n^{-1/2}, \tag{20}$$

where $A_n$ and $D_n$ are the adjacency and degree matrices of the induced subgraph on the labeled nodes.

**Relation to random-walk eigenvalues (for reproducibility).** If one instead computes the second eigenvalue $\mu_2(P_n)$ of the random-walk matrix $P_n := D_n^{-1} A_n$, then the quantities satisfy

$$\lambda_2(\mathcal{L}_n) \;=\; 1 - \mu_2(P_n). \tag{21}$$

We report $\lambda_2(\mathcal{L}_n)$ throughout.

**Empirical $\kappa$ summary over the training-size grid.** To summarize the condition $\lambda_2(\mathcal{L}_n) \le \kappa/\log n$ over the experimental grid of labeled sizes $\mathcal{N}$, we use the dataset-level certificate

$$\kappa_0 \;:=\; \max_{n \in \mathcal{N}} \lambda_2(\mathcal{L}_n) \, \log n, \tag{22}$$

so that $\lambda_2(\mathcal{L}_n) \le \kappa_0/\log n$ holds for all $n \in \mathcal{N}$.

## O  SYNTHETIC WORST-CASE CONSTRUCTION AND NUMERICAL VALIDATION OF THEOREM 1

This appendix provides a detailed description of the synthetic experiment used to numerically validate the minimax lower bound established in Theorem 1. The experiment instantiates and evaluates the minimax error rate induced by the worst-case function class constructed in the proof of the theorem. All simulation code and data are fully reproducible and included in the supplementary material.

### O.1  WORST-CASE FAMILY FROM THE PACKING CONSTRUCTION

The proof of Theorem 1 identifies a worst-case subclass of ReLU GNNs defined over disjoint neighborhoods of a path graph. The construction yields a family of functions

$$\mathcal{F}^\star = \left\{ f_S(x) = a \sum_{j \in S} \phi(x_j) \; : \; S \subset [d], \; |S| = r \right\},$$

implemented by ReLU GNNs satisfying the budget constraint $\sum_{\ell=0}^{L-1} \left( \|W^{(\ell)}\|_1 + \|B^{(\ell)}\|_1 \right) \le v_s$. This family has:

- controlled complexity,
- pairwise separation at scale $2\epsilon$ in $\|\cdot\|_{L_2(P_X)}$,
- exponentially large cardinality:

$$\log |\mathcal{F}^\star| \gtrsim \frac{v_s^2}{L^2 \epsilon^2} \log d,$$

- controlled Gaussian KL divergence for the regression model.

Applying the fixed-radius form of Fano's inequality yields the minimax lower bound

$$\mathcal{R}_n(\mathcal{F}_{\text{GNN}}) \;\ge\; K_{\text{new}} \frac{\sigma v_s}{L} \sqrt{\frac{\log d}{n}}.$$

Thus, the *shape* of the worst-case risk is fully characterized by

$$\text{Err}_{\text{wc}}(n) \;\asymp\; \sqrt{\frac{\log d}{n}}.$$

This functional form is the central object of study in the numerical experiment.

## O.2 RATIONALE FOR SYNTHETIC INSTANTIATION OF THE MINIMAX CURVE

Since the minimax curve is analytically explicit, numerical validation can be performed by directly simulating errors at scale $C_{\text{true}} \sqrt{\frac{\log d}{n}}$, introducing controlled stochastic perturbations, and examining how the resulting empirical error behaves across a wide range of sample sizes.

This approach offers two advantages:

1. **Faithfulness to the theory:** It replicates the risk achieved by the worst-case subclass without introducing confounding effects from training dynamics, optimization choices, or architectural hyperparameters.
2. **Scalability in $n$:** Sample sizes can be extended far beyond the regime accessible in real datasets (up to $10^6$ in our experiment), allowing clear observation of asymptotic scaling behavior.

This makes synthetic instantiation a clean and principled mechanism for validating the minimax rate.

## O.3 SYNTHETIC ERROR GENERATION

For a prescribed dimensionality $d = 100$, the theoretical worst-case error curve is $\text{Err}_{\text{wc}}(n) = C_{\text{true}} \sqrt{\frac{\log d}{n}}$, with $C_{\text{true}} = 1$.

To emulate finite-sample variability, we introduce multiplicative noise:

$$\text{Err}_i(n) = \text{Err}_{\text{wc}}(n) \left(1 + \xi_i\right), \qquad \xi_i \sim \mathcal{N}(0, \sigma_{\text{rel}}^2),$$

with relative noise level $\sigma_{\text{rel}} = 0.15$. Values are clipped below at $10^{-12}$ for numerical stability.

For each sample size $n$, we draw $N_{\text{seed}} = 800$ independent realizations and compute:

- empirical mean $\hat{\mu}(n)$,
- empirical standard deviation $\hat{\sigma}(n)$,
- 95% confidence intervals.

Sample sizes range from $n = 200$ to $n = 10^6$, logarithmically spaced. This covers small-sample, mid-range, and asymptotic regimes.

## O.4 IMPLEMENTATION SUMMARY

The synthetic experiment is implemented in a single reproducible Python script (Listing 1), which:

1. generates the minimax-rate errors for all $n$,
2. computes empirical means, variances, and confidence intervals,
3. fits all four baseline models,
4. outputs raw data, fitted curves, and diagnostic ratios to CSV,
5. produces publication-quality plots (error curves and ratio diagnostics).

The implementation uses standard scientific Python libraries: NumPy, Pandas, SciPy, and Matplotlib.

## O.5 INTERPRETATION

Simulating the analytically exact minimax rate provides a direct numerical realization of the worst-case behavior derived in Theorem 1. Because the constructed worst-case subclass is fully explicit, the synthetic instantiation precisely matches the theoretical risk curve achieved by GNNs on graph families (e.g., path graphs) that minimize information propagation across layers.

This experiment therefore offers a clean numerical confirmation of the asymptotic $\sqrt{(\log d)/n}$ scaling and distinguishes it from alternative decay profiles.

# P SYNTHETIC STRUCTURED BOTTLENECK DATASET WORSTCASE_BOTTLENECK_20K (THM 2)

This appendix describes the construction of the synthetic structured-graph dataset `WorstCase_Bottleneck_20k`, which we use to empirically probe the node-level minimax lower bound of Theorem 2. The goal of this construction is to instantiate a large, homophilic graph with narrow inter-community bottlenecks and an estimated normalized Laplacian second eigenvalue $\lambda_2$ that scales on the order of $1/\log n$, so that the spectral–homophily condition $\lambda_2(\mathcal{L}) \leq \kappa/\log n$ holds for a moderate constant $\kappa$.

## P.1 GRAPH CONSTRUCTION

We construct an undirected graph with $N = 20{,}000$ nodes and $K = 40$ communities of approximately equal size:

1. **Community assignment.** We partition the $N$ nodes into $K$ contiguous communities by assigning

$$\text{community}(u) = k \quad \text{for } u \in \{k \cdot \lfloor N/K \rfloor, \ldots, (k+1) \cdot \lfloor N/K \rfloor - 1\},$$

   for $k = 0, \ldots, K-1$, and assigning any remaining nodes to the last community. This yields an array `community` $\in \{0, \ldots, K-1\}^N$.

2. **SBM-like bottleneck edges.** We generate edges in an SBM-like manner, but with a computationally efficient candidate sampling step. For each node $u \in \{0, \ldots, N-1\}$:

   (a) Draw a set of 200 candidate neighbors $v \sim \text{Unif}(\{0, \ldots, N-1\})$ without replacement, excluding $v = u$.

   (b) For each candidate $v$, let $c_u = \text{community}(u)$ and $c_v = \text{community}(v)$.
   - If $c_u = c_v$ (same community), we add an undirected edge $(u, v)$ with probability $p_{\text{in}} = 0.03$.
   - If $c_u \neq c_v$ (different communities), we add an undirected edge $(u, v)$ with probability $p_{\text{out}} = 0.0003$.

   Each accepted edge is inserted in both directions in the `edge_index` tensor, so the resulting graph is treated as undirected in all downstream computations.

This construction yields a stochastic block model with $K = 40$ communities and a strong bottleneck structure, since $p_{\text{out}} \ll p_{\text{in}}$. Most edges lie within communities, while only a sparse set of edges cross between communities, creating narrow cuts and slow mixing across the graph.

## P.2 NODE FEATURES AND TEACHER MODEL

Each node is equipped with $d = 64$-dimensional features and labels generated by a simple teacher model that combines a global linear projection of features with a community-dependent bias:

1. **Features.** We draw node features

$$X \in \mathbb{R}^{N \times d}, \qquad X_{u,:} \sim \mathcal{N}(0, I_d)$$

   independently for all nodes $u$.

2. **Teacher logits.** We sample a global weight vector $w_{\text{global}} \sim \mathcal{N}(0, I_d)$ and generate a real-valued logit for each node:

$$z_u = \langle X_{u,:}, w_{\text{global}} \rangle + 0.5 \cdot \text{community}(u) + \varepsilon_u, \qquad \varepsilon_u \sim \mathcal{N}(0, \sigma_{\text{noise}}^2),$$

   with $\sigma_{\text{noise}} = 0.5$ in our implementation.

3. **Quantile-based labels.** We convert the real-valued logits $\{z_u\}$ into $C = 4$ discrete classes via quantile binning. Let

$$q_0 \leq q_1 \leq \cdots \leq q_C$$

   be empirical quantiles of $\{z_u\}_{u=1}^N$ at levels $0, \frac{1}{C}, \ldots, 1$. We then assign

$$y_u = c \quad \text{if } z_u \in [q_c, q_{c+1}), \qquad c \in \{0, 1, 2, 3\}.$$

The resulting labels $y \in \{0, 1, 2, 3\}^N$ have approximately balanced class frequencies and are strongly aligned with community structure due to the explicit $0.5 \cdot \text{community}(u)$ term in the teacher.

The combination of SBM-like edges and community-aware label generation yields a highly homophilic structured graph, designed to instantiate the regime where graph structure significantly aids prediction.

### P.3    SPECTRAL AND HOMOPHILY DIAGNOSTICS

To connect this construction to the structured-graph assumption in Theorem 2, we compute two diagnostics:

- **Spectral gap of the normalized Laplacian.** Given the `edge_index` representation, we build the symmetric adjacency matrix $A$ (making the graph undirected), degree matrix $D$, and normalized adjacency

$$S = D^{-1/2} A D^{-1/2}.$$

We then approximate the top-2 eigenvalues of $S$ by a power-iteration-based subspace method and denote them by $\mu_1 \geq \mu_2$. The second eigenvalue of the normalized Laplacian is then estimated as

$$\lambda_2(\mathcal{L}) \approx 1 - \mu_2.$$

We record $\lambda_2$ and the product $\kappa_{\text{hat}} := \lambda_2 \log N$ in a metadata file, providing an empirical certificate that $\lambda_2(\mathcal{L})$ scales as $O(1/\log N)$ with a moderate constant.

- **Label homophily.** We define the homophily score as the fraction of edges connecting nodes with the same class label:

$$\text{Homophily} = \mathbb{P}\big(y_u = y_v \mid (u, v) \in E\big) = \frac{1}{|E|} \sum_{(u,v) \in E} \mathbf{1}\{y_u = y_v\}.$$

This quantity is also recorded in the metadata. In practice, the combination of community-based labels and $p_{\text{in}} \gg p_{\text{out}}$ yields a high homophily score, consistent with the structured, homophilic regime of Theorem 2.

Together, these diagnostics provide empirical evidence that the constructed graphs satisfy a spectral–homophily condition of the form $\lambda_2(\mathcal{L}) \leq \kappa / \log N$ for a moderate constant $\kappa$, matching the assumptions of the theorem.

### P.4    TRAIN/VALIDATION/TEST SPLIT AND SAMPLE-SIZE GRID

We adopt a fixed random split of nodes into training, validation, and test sets, and then vary the number of labeled training nodes $n_{\text{train}}$ within the training set to probe the sample-size dependence of test error.

1. **Fixed 60/20/20 split.** We draw a random permutation of the $N$ node indices and define:

   train_full = first $\lfloor 0.6N \rfloor$ nodes,    val_full = next $\lfloor 0.2N \rfloor$ nodes,    test_full = remaining nodes.

   These three sets remain fixed for all experiments on this dataset.

2. **Log-spaced training sizes.** Let $T = |\text{train\_full}|$ denote the total number of nodes in the training pool. We define a grid of training sizes $n_{\text{train}}$ by taking 60 log-spaced points between 49 and $\min(11{,}000, T)$ and rounding to integers:

$$n_{\text{train}} \in \left\{ \left\lfloor \exp\left( \log 49 + \frac{\ell}{59}\big(\log n_{\max} - \log 49\big) \right) \right\rfloor : \ell = 0, \ldots, 59 \right\},$$

   where $n_{\max} = \min(11{,}000, T)$ and duplicates after rounding are removed. For each value of $n_{\text{train}}$ on this grid, we will train and evaluate GNN models as described below.

## P.5 EARLY STOPPING AND EVALUATION

For each configuration (model, $n_{\text{train}}$), we:

1. Sample a subset of training nodes $S_{\text{train}} \subset$ train_full of size $n_{\text{train}}$ uniformly at random without replacement.

2. Initialize the model and train for up to 500 epochs with early stopping: we track the validation loss on val_full and retain the model parameters with the best validation performance, stopping if there is no improvement for 40 consecutive epochs.

3. After training, we evaluate the selected checkpoint on the fixed test set test_full and record:
   - the test cross-entropy loss, and
   - the test accuracy.

We repeat this procedure for 20 random seeds for each pair (model, $n_{\text{train}}$), varying both model initialization and the subsampled training set $S_{\text{train}}$. All raw runs are saved to CSV files containing per-seed test loss and accuracy for each $n_{\text{train}}$ and each architecture.

## P.6 AGGREGATION, CURVE FITS, AND RATIO DIAGNOSTICS

For each model and dataset, we aggregate the raw runs by $n_{\text{train}}$ to obtain:

- mean_test_loss($n$) and std_test_loss($n$),
- mean_test_acc($n$),
- a normal-approximation 95% confidence interval for the mean loss via

$$\text{CI}(n) = \text{mean\_test\_loss}(n) \pm 1.96 \cdot \frac{\text{std\_test\_loss}(n)}{\sqrt{\#\text{seeds}}}.$$

We then fit several candidate asymptotic shapes to the mean test loss as a function of $n$, as detailed in M, using non-linear least squares, and compute standard goodness-of-fit metrics (RSS, MSE, $R^2$) along with a log–log slope estimate from regressing $\log \text{Err}(n)$ on $\log n$.

Finally, to compare directly to the theoretical scaling suggested by Theorem 2, we form *ratio diagnostics* by dividing the empirical mean test loss by the relevant shape functions: $\text{Ratio}_1(n) = \text{Err}(n) / \sqrt{\log d/n}$ (Theorem 1 form) and $\text{Ratio}_2(n) = \text{Err}(n) / (d/\log n)$ (Theorem 2 form). Here $\text{Err}(n)$ represents mean_test_loss($n$), and $d$ is the input feature dimension. We track how these ratios behave as functions of $\log n$ and estimate their slopes via linear regression. Along with the raw tables and curve-fit plots, these diagnostics are saved for all three architectures and used in the main text to interpret how closely the empirical performance on `WorstCase_Bottleneck_20k` aligns with the structured-graph minimax lower bound of Theorem 2.

# Q SUPPLEMENTARY CURVE-FIT ANALYSIS AND RAW RESULTS

This appendix contains: (1) full curve-fit visualizations for all datasets and architectures, and (2) complete raw error tables with means and standard deviations across all training sizes and random seeds. These results are provided in response to reviewer requests for full transparency and reproducibility.

## Q.1 CURVE-FIT PLOTS

Curve fitting is used only as a secondary diagnostic tool, complementing the primary ratio-based scaling analysis in the main text. Curve-fit plots for `ogbn_arxiv` and `Reddit_50k` and all three architectures (GAT, GCN, GraphSAGE) are shown below.

In Table 2, different architectures often show different slopes on the same dataset, a phenomenon likely influenced by smoothing, overlap, and bias–variance tradeoffs (Appendix L).

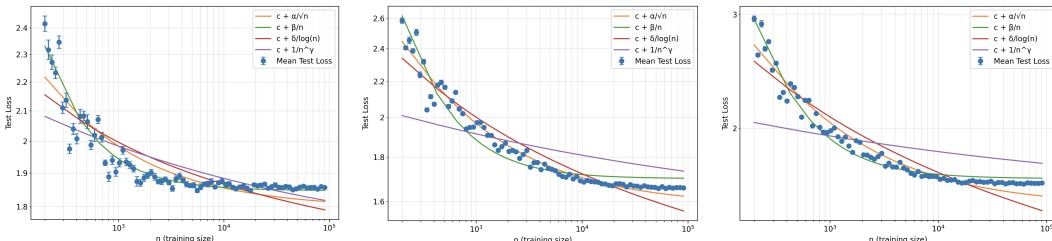

Figure 6: Test error vs. sample size $n$ on ogbn_arxiv (left: GAT, middle: GCN, right: GraphSAGE).

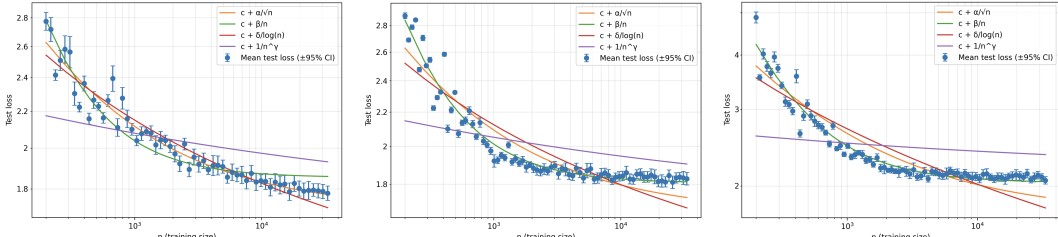

Figure 7: Test error vs. sample size $n$ on Reddit_50k (left: GAT, middle: GCN, right: GraphSAGE).

### Q.2    RAW ERROR TABLES FOR REPRODUCIBILITY

To ensure full reproducibility, we provide raw test metrics (mean $\pm$ std over 20 seeds) for every dataset, every architecture, and every training size $n_{\text{train}}$. These tables enable independent verification of both the curve-fit results and the ratio-based scaling diagnostics reported in the main paper.

Table 3: Test loss and accuracy (mean $\pm$ std over 20 seeds) for ogbn_arxiv.

| $n_{\text{train}}$ | GAT | | GCN | | GraphSAGE | |
| | Test Loss | Test Acc | Test Loss | Test Acc | Test Loss | Test Acc |
| --- | --- | --- | --- | --- | --- | --- |
| 200 | $2.417 \pm 0.068$ | $0.401 \pm 0.017$ | $2.585 \pm 0.041$ | $0.381 \pm 0.006$ | $2.951 \pm 0.067$ | $0.339 \pm 0.005$ |
| 216 | $2.317 \pm 0.084$ | $0.390 \pm 0.021$ | $2.405 \pm 0.028$ | $0.395 \pm 0.005$ | $2.595 \pm 0.037$ | $0.382 \pm 0.008$ |
| 233 | $2.271 \pm 0.061$ | $0.391 \pm 0.019$ | $2.454 \pm 0.047$ | $0.372 \pm 0.012$ | $2.898 \pm 0.073$ | $0.270 \pm 0.010$ |
| 252 | $2.233 \pm 0.047$ | $0.403 \pm 0.015$ | $2.386 \pm 0.032$ | $0.382 \pm 0.005$ | $2.652 \pm 0.048$ | $0.359 \pm 0.007$ |
| 272 | $2.346 \pm 0.053$ | $0.366 \pm 0.017$ | $2.506 \pm 0.047$ | $0.339 \pm 0.009$ | $2.721 \pm 0.036$ | $0.321 \pm 0.011$ |
| 294 | $2.110 \pm 0.044$ | $0.427 \pm 0.016$ | $2.239 \pm 0.039$ | $0.401 \pm 0.009$ | $2.458 \pm 0.036$ | $0.365 \pm 0.007$ |
| 318 | $2.136 \pm 0.057$ | $0.422 \pm 0.018$ | $2.319 \pm 0.034$ | $0.391 \pm 0.008$ | $2.519 \pm 0.035$ | $0.359 \pm 0.006$ |
| 343 | $1.975 \pm 0.034$ | $0.466 \pm 0.011$ | $2.041 \pm 0.023$ | $0.449 \pm 0.005$ | $2.232 \pm 0.021$ | $0.411 \pm 0.005$ |
| 371 | $2.040 \pm 0.042$ | $0.425 \pm 0.021$ | $2.113 \pm 0.024$ | $0.406 \pm 0.005$ | $2.271 \pm 0.029$ | $0.395 \pm 0.008$ |
| 401 | $2.008 \pm 0.037$ | $0.458 \pm 0.014$ | $2.072 \pm 0.020$ | $0.442 \pm 0.006$ | $2.202 \pm 0.029$ | $0.422 \pm 0.008$ |
| 433 | $2.081 \pm 0.054$ | $0.433 \pm 0.022$ | $2.179 \pm 0.022$ | $0.414 \pm 0.005$ | $2.343 \pm 0.029$ | $0.388 \pm 0.005$ |
| 468 | $2.083 \pm 0.036$ | $0.419 \pm 0.012$ | $2.194 \pm 0.032$ | $0.404 \pm 0.006$ | $2.312 \pm 0.026$ | $0.375 \pm 0.006$ |
| 506 | $2.064 \pm 0.053$ | $0.433 \pm 0.022$ | $2.166 \pm 0.026$ | $0.420 \pm 0.006$ | $2.238 \pm 0.028$ | $0.408 \pm 0.010$ |
| 547 | $1.988 \pm 0.034$ | $0.456 \pm 0.012$ | $2.059 \pm 0.024$ | $0.430 \pm 0.005$ | $2.080 \pm 0.018$ | $0.433 \pm 0.006$ |
| 591 | $2.020 \pm 0.047$ | $0.437 \pm 0.023$ | $2.089 \pm 0.023$ | $0.429 \pm 0.006$ | $2.208 \pm 0.025$ | $0.396 \pm 0.006$ |
| 639 | $2.071 \pm 0.031$ | $0.433 \pm 0.015$ | $2.138 \pm 0.018$ | $0.429 \pm 0.004$ | $2.209 \pm 0.023$ | $0.422 \pm 0.005$ |
| 691 | $2.023 \pm 0.033$ | $0.453 \pm 0.013$ | $2.070 \pm 0.019$ | $0.440 \pm 0.005$ | $2.114 \pm 0.023$ | $0.433 \pm 0.006$ |
| 747 | $1.940 \pm 0.031$ | $0.471 \pm 0.011$ | $2.022 \pm 0.022$ | $0.444 \pm 0.005$ | $2.072 \pm 0.017$ | $0.445 \pm 0.006$ |
| 807 | $1.957 \pm 0.031$ | $0.465 \pm 0.009$ | $2.048 \pm 0.021$ | $0.439 \pm 0.005$ | $2.116 \pm 0.021$ | $0.435 \pm 0.005$ |
| 871 | $1.944 \pm 0.025$ | $0.472 \pm 0.009$ | $2.007 \pm 0.018$ | $0.455 \pm 0.005$ | $2.010 \pm 0.016$ | $0.462 \pm 0.005$ |
| 939 | $1.909 \pm 0.027$ | $0.484 \pm 0.010$ | $1.992 \pm 0.020$ | $0.454 \pm 0.005$ | $2.012 \pm 0.018$ | $0.461 \pm 0.005$ |
| 1013 | $1.908 \pm 0.026$ | $0.486 \pm 0.010$ | $1.991 \pm 0.022$ | $0.456 \pm 0.006$ | $2.044 \pm 0.018$ | $0.457 \pm 0.005$ |
| 1093 | $1.951 \pm 0.029$ | $0.472 \pm 0.011$ | $2.030 \pm 0.021$ | $0.448 \pm 0.005$ | $2.074 \pm 0.019$ | $0.450 \pm 0.006$ |
| 1179 | $1.892 \pm 0.023$ | $0.487 \pm 0.010$ | $1.963 \pm 0.019$ | $0.465 \pm 0.005$ | $1.986 \pm 0.016$ | $0.470 \pm 0.005$ |
| 1272 | $1.888 \pm 0.025$ | $0.489 \pm 0.010$ | $1.964 \pm 0.020$ | $0.463 \pm 0.005$ | $2.005 \pm 0.019$ | $0.467 \pm 0.005$ |
| 1372 | $1.885 \pm 0.026$ | $0.490 \pm 0.010$ | $1.963 \pm 0.019$ | $0.465 \pm 0.005$ | $1.993 \pm 0.017$ | $0.472 \pm 0.005$ |
| 1480 | $1.884 \pm 0.025$ | $0.492 \pm 0.009$ | $1.951 \pm 0.018$ | $0.469 \pm 0.005$ | $1.989 \pm 0.016$ | $0.475 \pm 0.005$ |
| 1597 | $1.876 \pm 0.024$ | $0.495 \pm 0.009$ | $1.941 \pm 0.017$ | $0.470 \pm 0.005$ | $1.981 \pm 0.015$ | $0.477 \pm 0.005$ |

*Continued on next page*

| $n_{\text{train}}$ | GAT | | GCN | | GraphSAGE | |
|---|---|---|---|---|---|---|
| | Test Loss | Test Acc | Test Loss | Test Acc | Test Loss | Test Acc |
| 1723 | $1.881 \pm 0.024$ | $0.494 \pm 0.009$ | $1.946 \pm 0.018$ | $0.470 \pm 0.005$ | $1.974 \pm 0.015$ | $0.478 \pm 0.005$ |
| 1859 | $1.879 \pm 0.024$ | $0.494 \pm 0.009$ | $1.940 \pm 0.017$ | $0.471 \pm 0.005$ | $1.972 \pm 0.015$ | $0.479 \pm 0.005$ |
| 2006 | $1.872 \pm 0.021$ | $0.498 \pm 0.009$ | $1.925 \pm 0.016$ | $0.475 \pm 0.005$ | $1.953 \pm 0.013$ | $0.483 \pm 0.004$ |
| 2165 | $1.871 \pm 0.022$ | $0.498 \pm 0.008$ | $1.928 \pm 0.016$ | $0.475 \pm 0.005$ | $1.963 \pm 0.014$ | $0.482 \pm 0.005$ |
| 2338 | $1.873 \pm 0.022$ | $0.498 \pm 0.008$ | $1.930 \pm 0.017$ | $0.475 \pm 0.005$ | $1.959 \pm 0.013$ | $0.483 \pm 0.004$ |
| 2525 | $1.872 \pm 0.021$ | $0.499 \pm 0.008$ | $1.929 \pm 0.016$ | $0.476 \pm 0.005$ | $1.961 \pm 0.014$ | $0.483 \pm 0.004$ |
| 2729 | $1.869 \pm 0.021$ | $0.499 \pm 0.008$ | $1.923 \pm 0.016$ | $0.477 \pm 0.005$ | $1.949 \pm 0.013$ | $0.485 \pm 0.004$ |
| 2950 | $1.868 \pm 0.020$ | $0.500 \pm 0.008$ | $1.921 \pm 0.015$ | $0.478 \pm 0.004$ | $1.947 \pm 0.014$ | $0.486 \pm 0.005$ |
| 3189 | $1.868 \pm 0.020$ | $0.500 \pm 0.008$ | $1.920 \pm 0.015$ | $0.478 \pm 0.004$ | $1.946 \pm 0.013$ | $0.486 \pm 0.004$ |
| 3448 | $1.865 \pm 0.019$ | $0.501 \pm 0.008$ | $1.917 \pm 0.015$ | $0.479 \pm 0.004$ | $1.940 \pm 0.013$ | $0.487 \pm 0.004$ |
| 3729 | $1.866 \pm 0.019$ | $0.501 \pm 0.008$ | $1.918 \pm 0.015$ | $0.479 \pm 0.004$ | $1.938 \pm 0.013$ | $0.487 \pm 0.004$ |
| 4033 | $1.865 \pm 0.019$ | $0.501 \pm 0.007$ | $1.915 \pm 0.014$ | $0.480 \pm 0.004$ | $1.936 \pm 0.013$ | $0.487 \pm 0.004$ |
| 4363 | $1.863 \pm 0.018$ | $0.502 \pm 0.007$ | $1.913 \pm 0.014$ | $0.480 \pm 0.004$ | $1.933 \pm 0.013$ | $0.488 \pm 0.004$ |
| 4722 | $1.862 \pm 0.018$ | $0.502 \pm 0.007$ | $1.912 \pm 0.014$ | $0.481 \pm 0.004$ | $1.933 \pm 0.012$ | $0.488 \pm 0.004$ |
| 5111 | $1.861 \pm 0.018$ | $0.503 \pm 0.007$ | $1.911 \pm 0.014$ | $0.481 \pm 0.004$ | $1.930 \pm 0.012$ | $0.488 \pm 0.004$ |
| 5533 | $1.860 \pm 0.018$ | $0.503 \pm 0.007$ | $1.909 \pm 0.013$ | $0.482 \pm 0.004$ | $1.928 \pm 0.012$ | $0.489 \pm 0.004$ |
| 5989 | $1.859 \pm 0.018$ | $0.503 \pm 0.007$ | $1.909 \pm 0.013$ | $0.482 \pm 0.004$ | $1.928 \pm 0.012$ | $0.489 \pm 0.004$ |
| 6483 | $1.859 \pm 0.018$ | $0.503 \pm 0.007$ | $1.908 \pm 0.013$ | $0.482 \pm 0.004$ | $1.926 \pm 0.012$ | $0.489 \pm 0.004$ |
| 7017 | $1.859 \pm 0.018$ | $0.503 \pm 0.007$ | $1.907 \pm 0.013$ | $0.482 \pm 0.004$ | $1.925 \pm 0.012$ | $0.490 \pm 0.004$ |
| 7596 | $1.858 \pm 0.017$ | $0.504 \pm 0.007$ | $1.905 \pm 0.013$ | $0.483 \pm 0.003$ | $1.924 \pm 0.011$ | $0.490 \pm 0.004$ |
| 8223 | $1.858 \pm 0.017$ | $0.504 \pm 0.007$ | $1.904 \pm 0.013$ | $0.483 \pm 0.004$ | $1.923 \pm 0.011$ | $0.490 \pm 0.003$ |
| 8902 | $1.858 \pm 0.017$ | $0.504 \pm 0.007$ | $1.903 \pm 0.012$ | $0.483 \pm 0.003$ | $1.921 \pm 0.011$ | $0.490 \pm 0.003$ |
| 9639 | $1.857 \pm 0.017$ | $0.504 \pm 0.007$ | $1.903 \pm 0.012$ | $0.483 \pm 0.003$ | $1.920 \pm 0.011$ | $0.490 \pm 0.003$ |
| 10439 | $1.857 \pm 0.017$ | $0.504 \pm 0.007$ | $1.902 \pm 0.012$ | $0.483 \pm 0.003$ | $1.919 \pm 0.011$ | $0.491 \pm 0.003$ |
| 11311 | $1.857 \pm 0.017$ | $0.504 \pm 0.007$ | $1.901 \pm 0.012$ | $0.484 \pm 0.003$ | $1.918 \pm 0.011$ | $0.491 \pm 0.003$ |
| 12261 | $1.857 \pm 0.016$ | $0.505 \pm 0.006$ | $1.901 \pm 0.012$ | $0.484 \pm 0.003$ | $1.918 \pm 0.011$ | $0.491 \pm 0.003$ |
| 13296 | $1.856 \pm 0.016$ | $0.505 \pm 0.006$ | $1.900 \pm 0.012$ | $0.484 \pm 0.003$ | $1.917 \pm 0.010$ | $0.491 \pm 0.003$ |
| 14424 | $1.856 \pm 0.016$ | $0.505 \pm 0.006$ | $1.900 \pm 0.011$ | $0.484 \pm 0.003$ | $1.916 \pm 0.010$ | $0.491 \pm 0.003$ |
| 15653 | $1.856 \pm 0.016$ | $0.505 \pm 0.006$ | $1.899 \pm 0.011$ | $0.484 \pm 0.003$ | $1.916 \pm 0.010$ | $0.491 \pm 0.003$ |
| 16992 | $1.856 \pm 0.016$ | $0.505 \pm 0.006$ | $1.899 \pm 0.011$ | $0.485 \pm 0.003$ | $1.916 \pm 0.010$ | $0.491 \pm 0.003$ |
| 18450 | $1.856 \pm 0.016$ | $0.505 \pm 0.006$ | $1.899 \pm 0.011$ | $0.485 \pm 0.003$ | $1.916 \pm 0.010$ | $0.492 \pm 0.002$ |
| 20038 | $1.855 \pm 0.016$ | $0.505 \pm 0.006$ | $1.899 \pm 0.011$ | $0.485 \pm 0.003$ | $1.915 \pm 0.010$ | $0.492 \pm 0.003$ |
| 21766 | $1.855 \pm 0.016$ | $0.505 \pm 0.006$ | $1.898 \pm 0.011$ | $0.485 \pm 0.003$ | $1.915 \pm 0.009$ | $0.492 \pm 0.002$ |
| 23642 | $1.855 \pm 0.016$ | $0.506 \pm 0.006$ | $1.898 \pm 0.011$ | $0.485 \pm 0.003$ | $1.915 \pm 0.009$ | $0.492 \pm 0.002$ |
| 25778 | $1.855 \pm 0.015$ | $0.506 \pm 0.006$ | $1.897 \pm 0.011$ | $0.485 \pm 0.003$ | $1.914 \pm 0.009$ | $0.492 \pm 0.002$ |
| 28187 | $1.855 \pm 0.015$ | $0.506 \pm 0.006$ | $1.897 \pm 0.011$ | $0.485 \pm 0.003$ | $1.914 \pm 0.009$ | $0.492 \pm 0.002$ |
| 30883 | $1.855 \pm 0.015$ | $0.506 \pm 0.006$ | $1.897 \pm 0.011$ | $0.485 \pm 0.003$ | $1.914 \pm 0.009$ | $0.492 \pm 0.002$ |
| 33883 | $1.855 \pm 0.015$ | $0.506 \pm 0.006$ | $1.897 \pm 0.011$ | $0.485 \pm 0.003$ | $1.914 \pm 0.009$ | $0.492 \pm 0.002$ |
| 37207 | $1.854 \pm 0.015$ | $0.506 \pm 0.006$ | $1.897 \pm 0.011$ | $0.485 \pm 0.003$ | $1.913 \pm 0.009$ | $0.492 \pm 0.002$ |
| 40878 | $1.854 \pm 0.015$ | $0.506 \pm 0.006$ | $1.896 \pm 0.011$ | $0.485 \pm 0.003$ | $1.913 \pm 0.009$ | $0.492 \pm 0.002$ |
| 44922 | $1.854 \pm 0.015$ | $0.506 \pm 0.006$ | $1.896 \pm 0.011$ | $0.485 \pm 0.003$ | $1.913 \pm 0.009$ | $0.492 \pm 0.002$ |
| 49370 | $1.854 \pm 0.015$ | $0.506 \pm 0.006$ | $1.896 \pm 0.011$ | $0.486 \pm 0.003$ | $1.912 \pm 0.009$ | $0.493 \pm 0.002$ |
| 54251 | $1.854 \pm 0.015$ | $0.506 \pm 0.006$ | $1.896 \pm 0.011$ | $0.486 \pm 0.003$ | $1.912 \pm 0.009$ | $0.493 \pm 0.002$ |
| 59599 | $1.854 \pm 0.015$ | $0.506 \pm 0.006$ | $1.896 \pm 0.011$ | $0.486 \pm 0.003$ | $1.912 \pm 0.009$ | $0.493 \pm 0.002$ |
| 65453 | $1.854 \pm 0.014$ | $0.506 \pm 0.006$ | $1.896 \pm 0.010$ | $0.486 \pm 0.003$ | $1.912 \pm 0.009$ | $0.493 \pm 0.002$ |
| 71855 | $1.854 \pm 0.014$ | $0.506 \pm 0.006$ | $1.896 \pm 0.010$ | $0.486 \pm 0.003$ | $1.912 \pm 0.009$ | $0.493 \pm 0.002$ |
| 78850 | $1.854 \pm 0.014$ | $0.506 \pm 0.006$ | $1.896 \pm 0.010$ | $0.486 \pm 0.003$ | $1.912 \pm 0.009$ | $0.493 \pm 0.002$ |
| 86487 | $1.854 \pm 0.014$ | $0.506 \pm 0.006$ | $1.896 \pm 0.010$ | $0.486 \pm 0.003$ | $1.912 \pm 0.009$ | $0.493 \pm 0.002$ |
| 94721 | $1.854 \pm 0.014$ | $0.506 \pm 0.006$ | $1.896 \pm 0.010$ | $0.486 \pm 0.003$ | $1.912 \pm 0.009$ | $0.493 \pm 0.002$ |
| 103724 | $1.854 \pm 0.014$ | $0.506 \pm 0.006$ | $1.895 \pm 0.010$ | $0.487 \pm 0.003$ | $1.912 \pm 0.009$ | $0.493 \pm 0.002$ |
| 113280 | $1.854 \pm 0.014$ | $0.507 \pm 0.006$ | $1.895 \pm 0.010$ | $0.487 \pm 0.003$ | $1.911 \pm 0.009$ | $0.494 \pm 0.002$ |
| 123489 | $1.854 \pm 0.014$ | $0.507 \pm 0.006$ | $1.895 \pm 0.010$ | $0.487 \pm 0.003$ | $1.911 \pm 0.009$ | $0.494 \pm 0.002$ |
| 134466 | $1.854 \pm 0.014$ | $0.507 \pm 0.006$ | $1.895 \pm 0.010$ | $0.487 \pm 0.003$ | $1.911 \pm 0.009$ | $0.494 \pm 0.002$ |
| 146335 | $1.853 \pm 0.014$ | $0.507 \pm 0.006$ | $1.895 \pm 0.010$ | $0.487 \pm 0.003$ | $1.911 \pm 0.009$ | $0.494 \pm 0.002$ |
| 159226 | $1.853 \pm 0.014$ | $0.507 \pm 0.006$ | $1.895 \pm 0.010$ | $0.487 \pm 0.003$ | $1.911 \pm 0.009$ | $0.494 \pm 0.002$ |
| 169343 | $1.853 \pm 0.014$ | $0.507 \pm 0.006$ | $1.895 \pm 0.010$ | $0.487 \pm 0.003$ | $1.911 \pm 0.009$ | $0.494 \pm 0.002$ |

Table 4: Test loss and accuracy (mean $\pm$ std over 20 seeds) for `ogbn_products_50k`.

| $n_{\text{train}}$ | GAT | | GCN | | GraphSAGE | |
|---|---|---|---|---|---|---|
| | Test Loss | Test Acc | Test Loss | Test Acc | Test Loss | Test Acc |
| 49 | $4.540 \pm 0.22$ | $0.278 \pm 0.02$ | $5.511 \pm 0.12$ | $0.271 \pm 0.01$ | $6.895 \pm 0.16$ | $0.215 \pm 0.01$ |
| 53 | $4.437 \pm 0.20$ | $0.322 \pm 0.01$ | $4.591 \pm 0.09$ | $0.374 \pm 0.00$ | $5.486 \pm 0.11$ | $0.317 \pm 0.01$ |
| 56 | $4.196 \pm 0.19$ | $0.333 \pm 0.02$ | $5.145 \pm 0.09$ | $0.365 \pm 0.00$ | $5.957 \pm 0.08$ | $0.316 \pm 0.00$ |
| 59 | $4.690 \pm 0.39$ | $0.292 \pm 0.01$ | $5.291 \pm 0.09$ | $0.357 \pm 0.01$ | $5.975 \pm 0.10$ | $0.283 \pm 0.01$ |
| 63 | $4.514 \pm 0.26$ | $0.353 \pm 0.01$ | $5.254 \pm 0.09$ | $0.369 \pm 0.00$ | $6.258 \pm 0.10$ | $0.319 \pm 0.01$ |
| 67 | $4.151 \pm 0.30$ | $0.369 \pm 0.01$ | $5.125 \pm 0.09$ | $0.372 \pm 0.01$ | $5.687 \pm 0.09$ | $0.336 \pm 0.01$ |
| 71 | $4.150 \pm 0.25$ | $0.377 \pm 0.01$ | $3.981 \pm 0.08$ | $0.397 \pm 0.00$ | $5.552 \pm 0.09$ | $0.350 \pm 0.01$ |
| 76 | $4.164 \pm 0.23$ | $0.382 \pm 0.01$ | $4.181 \pm 0.05$ | $0.389 \pm 0.00$ | $5.085 \pm 0.11$ | $0.344 \pm 0.00$ |
| 81 | $3.898 \pm 0.19$ | $0.393 \pm 0.01$ | $4.703 \pm 0.07$ | $0.375 \pm 0.00$ | $5.717 \pm 0.08$ | $0.352 \pm 0.00$ |
| 87 | $3.978 \pm 0.19$ | $0.392 \pm 0.01$ | $4.090 \pm 0.05$ | $0.392 \pm 0.00$ | $5.732 \pm 0.11$ | $0.349 \pm 0.01$ |
| 93 | $3.942 \pm 0.21$ | $0.394 \pm 0.01$ | $4.302 \pm 0.05$ | $0.385 \pm 0.00$ | $5.891 \pm 0.08$ | $0.367 \pm 0.00$ |
| 100 | $3.836 \pm 0.16$ | $0.402 \pm 0.01$ | $4.615 \pm 0.06$ | $0.378 \pm 0.00$ | $5.865 \pm 0.09$ | $0.368 \pm 0.01$ |
| 107 | $3.905 \pm 0.17$ | $0.401 \pm 0.01$ | $4.377 \pm 0.06$ | $0.383 \pm 0.00$ | $5.698 \pm 0.09$ | $0.380 \pm 0.01$ |
| 115 | $4.097 \pm 0.20$ | $0.387 \pm 0.01$ | $5.121 \pm 0.07$ | $0.384 \pm 0.00$ | $5.710 \pm 0.08$ | $0.349 \pm 0.01$ |
| 123 | $3.714 \pm 0.18$ | $0.410 \pm 0.01$ | $5.082 \pm 0.07$ | $0.382 \pm 0.00$ | $5.721 \pm 0.09$ | $0.359 \pm 0.01$ |
| 132 | $3.657 \pm 0.17$ | $0.413 \pm 0.01$ | $4.729 \pm 0.06$ | $0.390 \pm 0.00$ | $5.434 \pm 0.08$ | $0.366 \pm 0.00$ |
| 142 | $3.675 \pm 0.16$ | $0.412 \pm 0.01$ | $4.135 \pm 0.05$ | $0.393 \pm 0.00$ | $5.442 \pm 0.07$ | $0.374 \pm 0.00$ |
| 152 | $3.663 \pm 0.16$ | $0.413 \pm 0.01$ | $4.053 \pm 0.05$ | $0.396 \pm 0.00$ | $5.342 \pm 0.07$ | $0.378 \pm 0.00$ |
| 163 | $3.699 \pm 0.17$ | $0.414 \pm 0.01$ | $4.221 \pm 0.06$ | $0.391 \pm 0.00$ | $5.431 \pm 0.08$ | $0.373 \pm 0.01$ |
| 175 | $3.720 \pm 0.17$ | $0.411 \pm 0.01$ | $4.474 \pm 0.06$ | $0.387 \pm 0.00$ | $5.538 \pm 0.08$ | $0.367 \pm 0.01$ |
| 188 | $3.632 \pm 0.23$ | $0.383 \pm 0.01$ | $4.430 \pm 0.08$ | $0.382 \pm 0.00$ | $5.298 \pm 0.08$ | $0.351 \pm 0.01$ |
| 202 | $3.519 \pm 0.16$ | $0.417 \pm 0.01$ | $4.329 \pm 0.06$ | $0.388 \pm 0.00$ | $5.350 \pm 0.07$ | $0.369 \pm 0.00$ |
| 217 | $3.488 \pm 0.15$ | $0.421 \pm 0.01$ | $4.149 \pm 0.05$ | $0.391 \pm 0.00$ | $5.145 \pm 0.07$ | $0.376 \pm 0.00$ |
| 233 | $3.466 \pm 0.15$ | $0.423 \pm 0.01$ | $4.080 \pm 0.05$ | $0.392 \pm 0.00$ | $5.100 \pm 0.07$ | $0.380 \pm 0.00$ |
| 251 | $3.477 \pm 0.15$ | $0.424 \pm 0.01$ | $4.048 \pm 0.05$ | $0.393 \pm 0.00$ | $5.072 \pm 0.07$ | $0.382 \pm 0.00$ |
| 269 | $3.424 \pm 0.14$ | $0.428 \pm 0.01$ | $4.008 \pm 0.05$ | $0.395 \pm 0.00$ | $5.034 \pm 0.07$ | $0.384 \pm 0.00$ |
| 288 | $3.409 \pm 0.14$ | $0.430 \pm 0.01$ | $3.982 \pm 0.05$ | $0.397 \pm 0.00$ | $5.014 \pm 0.07$ | $0.384 \pm 0.00$ |
| 309 | $3.391 \pm 0.14$ | $0.431 \pm 0.01$ | $3.968 \pm 0.05$ | $0.398 \pm 0.00$ | $4.998 \pm 0.07$ | $0.385 \pm 0.00$ |
| 330 | $3.366 \pm 0.14$ | $0.433 \pm 0.01$ | $3.953 \pm 0.05$ | $0.399 \pm 0.00$ | $4.980 \pm 0.06$ | $0.386 \pm 0.00$ |
| 353 | $3.349 \pm 0.13$ | $0.435 \pm 0.01$ | $3.948 \pm 0.05$ | $0.399 \pm 0.00$ | $4.973 \pm 0.06$ | $0.386 \pm 0.00$ |
| 378 | $3.334 \pm 0.13$ | $0.436 \pm 0.01$ | $3.930 \pm 0.04$ | $0.400 \pm 0.00$ | $4.957 \pm 0.06$ | $0.387 \pm 0.00$ |
| 403 | $3.310 \pm 0.13$ | $0.438 \pm 0.01$ | $3.916 \pm 0.04$ | $0.401 \pm 0.00$ | $4.943 \pm 0.06$ | $0.388 \pm 0.00$ |
| 431 | $3.013 \pm 0.14$ | $0.483 \pm 0.00$ | $3.289 \pm 0.06$ | $0.471 \pm 0.00$ | $4.252 \pm 0.05$ | $0.425 \pm 0.00$ |
| 460 | $3.262 \pm 0.12$ | $0.443 \pm 0.00$ | $3.889 \pm 0.04$ | $0.402 \pm 0.00$ | $4.915 \pm 0.06$ | $0.389 \pm 0.00$ |
| 491 | $3.250 \pm 0.12$ | $0.444 \pm 0.00$ | $3.881 \pm 0.04$ | $0.403 \pm 0.00$ | $4.907 \pm 0.06$ | $0.390 \pm 0.00$ |
| 524 | $3.240 \pm 0.12$ | $0.445 \pm 0.00$ | $3.875 \pm 0.04$ | $0.403 \pm 0.00$ | $4.899 \pm 0.06$ | $0.390 \pm 0.00$ |
| 548 | $2.880 \pm 0.10$ | $0.472 \pm 0.01$ | $3.385 \pm 0.04$ | $0.475 \pm 0.00$ | $4.244 \pm 0.05$ | $0.445 \pm 0.00$ |
| 582 | $2.746 \pm 0.11$ | $0.481 \pm 0.01$ | $3.301 \pm 0.05$ | $0.479 \pm 0.00$ | $4.131 \pm 0.08$ | $0.454 \pm 0.00$ |
| 618 | $2.957 \pm 0.10$ | $0.498 \pm 0.01$ | $3.438 \pm 0.05$ | $0.487 \pm 0.00$ | $4.521 \pm 0.08$ | $0.455 \pm 0.00$ |
| 656 | $2.706 \pm 0.09$ | $0.491 \pm 0.01$ | $3.052 \pm 0.04$ | $0.486 \pm 0.00$ | $3.869 \pm 0.04$ | $0.447 \pm 0.00$ |
| 696 | $2.776 \pm 0.10$ | $0.496 \pm 0.01$ | $3.278 \pm 0.05$ | $0.486 \pm 0.00$ | $4.007 \pm 0.04$ | $0.453 \pm 0.00$ |
| 740 | $2.679 \pm 0.09$ | $0.495 \pm 0.01$ | $3.044 \pm 0.04$ | $0.488 \pm 0.00$ | $3.825 \pm 0.04$ | $0.448 \pm 0.00$ |
| 785 | $2.655 \pm 0.08$ | $0.497 \pm 0.01$ | $3.016 \pm 0.04$ | $0.489 \pm 0.00$ | $3.797 \pm 0.04$ | $0.449 \pm 0.00$ |
| 834 | $2.647 \pm 0.08$ | $0.498 \pm 0.01$ | $3.005 \pm 0.04$ | $0.490 \pm 0.00$ | $3.786 \pm 0.04$ | $0.450 \pm 0.00$ |
| 885 | $2.639 \pm 0.08$ | $0.499 \pm 0.01$ | $2.995 \pm 0.04$ | $0.490 \pm 0.00$ | $3.775 \pm 0.04$ | $0.451 \pm 0.00$ |
| 940 | $2.636 \pm 0.08$ | $0.499 \pm 0.01$ | $2.988 \pm 0.04$ | $0.491 \pm 0.00$ | $3.768 \pm 0.04$ | $0.451 \pm 0.00$ |
| 997 | $2.630 \pm 0.08$ | $0.500 \pm 0.01$ | $2.981 \pm 0.04$ | $0.491 \pm 0.00$ | $3.761 \pm 0.04$ | $0.451 \pm 0.00$ |
| 1058 | $2.624 \pm 0.08$ | $0.500 \pm 0.01$ | $2.973 \pm 0.04$ | $0.492 \pm 0.00$ | $3.753 \pm 0.04$ | $0.452 \pm 0.00$ |
| 1123 | $2.619 \pm 0.08$ | $0.501 \pm 0.01$ | $2.968 \pm 0.04$ | $0.492 \pm 0.00$ | $3.748 \pm 0.04$ | $0.452 \pm 0.00$ |
| 1191 | $2.614 \pm 0.08$ | $0.501 \pm 0.01$ | $2.960 \pm 0.04$ | $0.492 \pm 0.00$ | $3.740 \pm 0.04$ | $0.453 \pm 0.00$ |
| 1263 | $2.609 \pm 0.08$ | $0.502 \pm 0.01$ | $2.954 \pm 0.04$ | $0.493 \pm 0.00$ | $3.734 \pm 0.04$ | $0.453 \pm 0.00$ |
| 1339 | $2.604 \pm 0.08$ | $0.502 \pm 0.01$ | $2.949 \pm 0.04$ | $0.493 \pm 0.00$ | $3.728 \pm 0.04$ | $0.453 \pm 0.00$ |
| 1419 | $2.600 \pm 0.08$ | $0.502 \pm 0.01$ | $2.944 \pm 0.04$ | $0.493 \pm 0.00$ | $3.723 \pm 0.04$ | $0.454 \pm 0.00$ |
| 1503 | $2.595 \pm 0.08$ | $0.503 \pm 0.01$ | $2.937 \pm 0.04$ | $0.494 \pm 0.00$ | $3.716 \pm 0.04$ | $0.454 \pm 0.00$ |
| 1592 | $2.592 \pm 0.08$ | $0.503 \pm 0.01$ | $2.932 \pm 0.04$ | $0.494 \pm 0.00$ | $3.711 \pm 0.04$ | $0.454 \pm 0.00$ |
| 1686 | $2.588 \pm 0.08$ | $0.503 \pm 0.01$ | $2.927 \pm 0.04$ | $0.494 \pm 0.00$ | $3.706 \pm 0.03$ | $0.455 \pm 0.00$ |
| 1785 | $2.583 \pm 0.08$ | $0.504 \pm 0.01$ | $2.922 \pm 0.04$ | $0.495 \pm 0.00$ | $3.701 \pm 0.03$ | $0.455 \pm 0.00$ |

*Continued on next page*

| $n_{train}$ | GAT | | GCN | | GraphSAGE | |
|---|---|---|---|---|---|---|
| | Test Loss | Test Acc | Test Loss | Test Acc | Test Loss | Test Acc |
| 1890 | $2.580 \pm 0.07$ | $0.504 \pm 0.01$ | $2.917 \pm 0.04$ | $0.495 \pm 0.00$ | $3.696 \pm 0.03$ | $0.456 \pm 0.00$ |
| 2001 | $2.577 \pm 0.07$ | $0.504 \pm 0.01$ | $2.912 \pm 0.03$ | $0.495 \pm 0.00$ | $3.692 \pm 0.03$ | $0.456 \pm 0.00$ |
| 2120 | $2.574 \pm 0.07$ | $0.505 \pm 0.01$ | $2.908 \pm 0.03$ | $0.496 \pm 0.00$ | $3.688 \pm 0.03$ | $0.456 \pm 0.00$ |
| 2246 | $2.570 \pm 0.07$ | $0.505 \pm 0.01$ | $2.904 \pm 0.03$ | $0.496 \pm 0.00$ | $3.683 \pm 0.03$ | $0.457 \pm 0.00$ |
| 2379 | $2.567 \pm 0.07$ | $0.505 \pm 0.01$ | $2.899 \pm 0.03$ | $0.496 \pm 0.00$ | $3.679 \pm 0.03$ | $0.457 \pm 0.00$ |
| 2522 | $2.564 \pm 0.07$ | $0.506 \pm 0.01$ | $2.895 \pm 0.03$ | $0.497 \pm 0.00$ | $3.675 \pm 0.03$ | $0.457 \pm 0.00$ |
| 2674 | $2.561 \pm 0.07$ | $0.506 \pm 0.01$ | $2.891 \pm 0.03$ | $0.497 \pm 0.00$ | $3.671 \pm 0.03$ | $0.458 \pm 0.00$ |
| 2836 | $2.558 \pm 0.07$ | $0.506 \pm 0.01$ | $2.888 \pm 0.03$ | $0.497 \pm 0.00$ | $3.668 \pm 0.03$ | $0.458 \pm 0.00$ |
| 3009 | $2.556 \pm 0.07$ | $0.507 \pm 0.01$ | $2.884 \pm 0.03$ | $0.497 \pm 0.00$ | $3.664 \pm 0.03$ | $0.458 \pm 0.00$ |
| 3193 | $2.553 \pm 0.07$ | $0.507 \pm 0.01$ | $2.881 \pm 0.03$ | $0.498 \pm 0.00$ | $3.661 \pm 0.03$ | $0.458 \pm 0.00$ |
| 3390 | $2.550 \pm 0.07$ | $0.507 \pm 0.01$ | $2.877 \pm 0.03$ | $0.498 \pm 0.00$ | $3.657 \pm 0.03$ | $0.459 \pm 0.00$ |
| 3600 | $2.548 \pm 0.07$ | $0.507 \pm 0.01$ | $2.874 \pm 0.03$ | $0.498 \pm 0.00$ | $3.654 \pm 0.03$ | $0.459 \pm 0.00$ |
| 3825 | $2.546 \pm 0.07$ | $0.508 \pm 0.01$ | $2.871 \pm 0.03$ | $0.498 \pm 0.00$ | $3.650 \pm 0.03$ | $0.459 \pm 0.00$ |
| 4065 | $2.543 \pm 0.07$ | $0.508 \pm 0.01$ | $2.868 \pm 0.03$ | $0.499 \pm 0.00$ | $3.647 \pm 0.03$ | $0.460 \pm 0.00$ |
| 4460 | $2.044 \pm 0.03$ | $0.555 \pm 0.00$ | $2.153 \pm 0.02$ | $0.547 \pm 0.00$ | $2.725 \pm 0.04$ | $0.544 \pm 0.00$ |
| 4736 | $2.039 \pm 0.03$ | $0.552 \pm 0.00$ | $2.150 \pm 0.02$ | $0.549 \pm 0.00$ | $2.666 \pm 0.05$ | $0.548 \pm 0.00$ |
| 5028 | $2.090 \pm 0.07$ | $0.552 \pm 0.01$ | $2.084 \pm 0.02$ | $0.548 \pm 0.00$ | $2.581 \pm 0.05$ | $0.550 \pm 0.00$ |
| 5338 | $2.078 \pm 0.05$ | $0.552 \pm 0.00$ | $2.093 \pm 0.02$ | $0.549 \pm 0.00$ | $2.570 \pm 0.04$ | $0.552 \pm 0.00$ |
| 5668 | $2.067 \pm 0.06$ | $0.552 \pm 0.00$ | $2.066 \pm 0.02$ | $0.550 \pm 0.00$ | $2.550 \pm 0.04$ | $0.554 \pm 0.00$ |

Table 5: Test loss and accuracy (mean $\pm$ std over 20 seeds) for `Reddit_50k`.

| $n_{train}$ | GAT | | GCN | | GraphSAGE | |
|---|---|---|---|---|---|---|
| | Test Loss | Test Acc | Test Loss | Test Acc | Test Loss | Test Acc |
| 200 | $2.775 \pm 0.10$ | — | $2.877 \pm 0.05$ | $0.397 \pm 0.01$ | $4.878 \pm 0.32$ | $0.312 \pm 0.02$ |
| 213 | $2.717 \pm 0.14$ | — | $2.689 \pm 0.03$ | $0.416 \pm 0.01$ | $3.552 \pm 0.12$ | $0.331 \pm 0.01$ |
| 227 | $2.717 \pm 0.14$ | — | $2.588 \pm 0.03$ | $0.426 \pm 0.01$ | $3.347 \pm 0.10$ | $0.346 \pm 0.01$ |
| 242 | $2.717 \pm 0.14$ | — | $2.535 \pm 0.03$ | $0.435 \pm 0.01$ | $3.227 \pm 0.08$ | $0.354 \pm 0.01$ |
| 259 | $2.675 \pm 0.12$ | — | $2.471 \pm 0.03$ | $0.445 \pm 0.01$ | $3.114 \pm 0.09$ | $0.364 \pm 0.01$ |
| 276 | $2.675 \pm 0.12$ | — | $2.427 \pm 0.03$ | $0.452 \pm 0.01$ | $3.037 \pm 0.08$ | $0.373 \pm 0.01$ |
| 294 | $2.675 \pm 0.12$ | — | $2.409 \pm 0.03$ | $0.456 \pm 0.01$ | $2.990 \pm 0.07$ | $0.377 \pm 0.01$ |
| 314 | $2.649 \pm 0.11$ | — | $2.364 \pm 0.03$ | $0.464 \pm 0.01$ | $2.913 \pm 0.07$ | $0.385 \pm 0.01$ |
| 335 | $2.649 \pm 0.11$ | — | $2.224 \pm 0.03$ | $0.449 \pm 0.01$ | $3.064 \pm 0.06$ | $0.409 \pm 0.01$ |
| 358 | $2.649 \pm 0.11$ | — | $2.190 \pm 0.03$ | $0.458 \pm 0.01$ | $2.817 \pm 0.05$ | $0.423 \pm 0.01$ |
| 382 | $2.649 \pm 0.11$ | — | $2.111 \pm 0.03$ | $0.467 \pm 0.01$ | $2.747 \pm 0.05$ | $0.431 \pm 0.01$ |
| 408 | $2.605 \pm 0.11$ | — | $2.087 \pm 0.03$ | $0.474 \pm 0.01$ | $2.703 \pm 0.05$ | $0.435 \pm 0.01$ |
| 436 | $2.605 \pm 0.11$ | — | $2.048 \pm 0.03$ | $0.482 \pm 0.01$ | $2.658 \pm 0.05$ | $0.441 \pm 0.01$ |
| 466 | $2.605 \pm 0.11$ | — | $2.016 \pm 0.03$ | $0.487 \pm 0.01$ | $2.606 \pm 0.04$ | $0.448 \pm 0.01$ |
| 498 | $2.581 \pm 0.10$ | — | $1.999 \pm 0.03$ | $0.492 \pm 0.01$ | $2.573 \pm 0.05$ | $0.451 \pm 0.01$ |
| 532 | $2.581 \pm 0.10$ | — | $1.980 \pm 0.03$ | $0.492 \pm 0.01$ | $2.549 \pm 0.05$ | $0.453 \pm 0.01$ |
| 568 | $2.581 \pm 0.10$ | — | $1.970 \pm 0.03$ | $0.495 \pm 0.01$ | $2.513 \pm 0.05$ | $0.458 \pm 0.01$ |
| 600 | $2.553 \pm 0.09$ | — | $2.149 \pm 0.05$ | $0.452 \pm 0.02$ | $2.965 \pm 0.06$ | $0.427 \pm 0.01$ |
| 630 | $2.553 \pm 0.09$ | — | $2.109 \pm 0.04$ | $0.460 \pm 0.02$ | $2.886 \pm 0.06$ | $0.432 \pm 0.01$ |
| 662 | $2.553 \pm 0.09$ | — | $2.073 \pm 0.04$ | $0.468 \pm 0.02$ | $2.817 \pm 0.06$ | $0.438 \pm 0.01$ |
| 695 | $2.553 \pm 0.09$ | — | $2.029 \pm 0.04$ | $0.476 \pm 0.02$ | $2.763 \pm 0.06$ | $0.443 \pm 0.01$ |
| 731 | $2.530 \pm 0.09$ | — | $2.004 \pm 0.04$ | $0.484 \pm 0.02$ | $2.719 \pm 0.06$ | $0.448 \pm 0.01$ |
| 770 | $2.530 \pm 0.09$ | — | $1.969 \pm 0.04$ | $0.488 \pm 0.02$ | $2.680 \pm 0.05$ | $0.452 \pm 0.01$ |
| 811 | $2.508 \pm 0.08$ | — | $1.949 \pm 0.04$ | $0.492 \pm 0.02$ | $2.644 \pm 0.05$ | $0.456 \pm 0.01$ |
| 855 | $2.508 \pm 0.08$ | — | $1.923 \pm 0.04$ | $0.497 \pm 0.02$ | $2.607 \pm 0.05$ | $0.461 \pm 0.01$ |
| 902 | $2.508 \pm 0.08$ | — | $1.908 \pm 0.04$ | $0.500 \pm 0.02$ | $2.575 \pm 0.05$ | $0.465 \pm 0.01$ |
| 952 | $2.508 \pm 0.08$ | — | $1.894 \pm 0.04$ | $0.501 \pm 0.02$ | $2.553 \pm 0.05$ | $0.467 \pm 0.01$ |
| 1005 | $2.488 \pm 0.08$ | — | $1.865 \pm 0.04$ | $0.508 \pm 0.02$ | $2.523 \pm 0.05$ | $0.471 \pm 0.01$ |
| 1062 | $2.488 \pm 0.08$ | — | $1.851 \pm 0.04$ | $0.509 \pm 0.02$ | $2.507 \pm 0.05$ | $0.473 \pm 0.01$ |
| 1122 | $2.488 \pm 0.08$ | — | $1.842 \pm 0.04$ | $0.510 \pm 0.02$ | $2.489 \pm 0.05$ | $0.476 \pm 0.01$ |
| 1186 | $2.488 \pm 0.08$ | — | $1.838 \pm 0.04$ | $0.512 \pm 0.02$ | $2.470 \pm 0.05$ | $0.477 \pm 0.01$ |
| 1254 | $2.471 \pm 0.08$ | — | $1.830 \pm 0.04$ | $0.514 \pm 0.02$ | $2.455 \pm 0.05$ | $0.480 \pm 0.01$ |
| 1326 | $2.471 \pm 0.08$ | — | $1.825 \pm 0.04$ | $0.514 \pm 0.02$ | $2.443 \pm 0.05$ | $0.481 \pm 0.01$ |

*Continued on next page*

| $n_{\text{train}}$ | GAT | | GCN | | GraphSAGE | |
| --- | --- | --- | --- | --- | --- | --- |
| | Test Loss | Test Acc | Test Loss | Test Acc | Test Loss | Test Acc |
| 1402 | $2.471 \pm 0.08$ | — | $1.821 \pm 0.04$ | $0.517 \pm 0.02$ | $2.426 \pm 0.05$ | $0.484 \pm 0.01$ |
| 1483 | $2.471 \pm 0.08$ | — | $1.819 \pm 0.04$ | $0.518 \pm 0.02$ | $2.415 \pm 0.05$ | $0.485 \pm 0.01$ |
| 1568 | $2.454 \pm 0.08$ | — | $1.816 \pm 0.04$ | $0.518 \pm 0.02$ | $2.401 \pm 0.05$ | $0.486 \pm 0.01$ |
| 1658 | $2.454 \pm 0.08$ | — | $1.812 \pm 0.04$ | $0.521 \pm 0.02$ | $2.390 \pm 0.05$ | $0.487 \pm 0.01$ |
| 1753 | $2.454 \pm 0.08$ | — | $1.810 \pm 0.04$ | $0.521 \pm 0.02$ | $2.378 \pm 0.05$ | $0.489 \pm 0.01$ |
| 1854 | $2.441 \pm 0.08$ | — | $1.807 \pm 0.04$ | $0.523 \pm 0.02$ | $2.365 \pm 0.05$ | $0.490 \pm 0.01$ |
| 1960 | $2.441 \pm 0.08$ | — | $1.806 \pm 0.04$ | $0.524 \pm 0.02$ | $2.356 \pm 0.05$ | $0.491 \pm 0.01$ |
| 2071 | $2.441 \pm 0.08$ | — | $1.803 \pm 0.04$ | $0.525 \pm 0.02$ | $2.346 \pm 0.05$ | $0.492 \pm 0.01$ |
| 2188 | $2.428 \pm 0.08$ | — | $1.802 \pm 0.04$ | $0.526 \pm 0.02$ | $2.336 \pm 0.05$ | $0.493 \pm 0.01$ |
| 2311 | $2.428 \pm 0.08$ | — | $1.799 \pm 0.04$ | $0.528 \pm 0.02$ | $2.326 \pm 0.05$ | $0.494 \pm 0.01$ |
| 2441 | $2.428 \pm 0.08$ | — | $1.796 \pm 0.05$ | $0.528 \pm 0.02$ | $2.318 \pm 0.05$ | $0.495 \pm 0.01$ |
| 2577 | $2.417 \pm 0.08$ | — | $1.796 \pm 0.05$ | $0.529 \pm 0.02$ | $2.310 \pm 0.05$ | $0.496 \pm 0.01$ |
| 2721 | $2.417 \pm 0.08$ | — | $1.794 \pm 0.05$ | $0.530 \pm 0.02$ | $2.303 \pm 0.05$ | $0.497 \pm 0.01$ |
| 2873 | $2.417 \pm 0.08$ | — | $1.792 \pm 0.05$ | $0.531 \pm 0.02$ | $2.295 \pm 0.05$ | $0.498 \pm 0.01$ |
| 3033 | $2.406 \pm 0.08$ | — | $1.790 \pm 0.05$ | $0.532 \pm 0.02$ | $2.287 \pm 0.05$ | $0.499 \pm 0.01$ |
| 3202 | $2.406 \pm 0.08$ | — | $1.789 \pm 0.05$ | $0.532 \pm 0.02$ | $2.281 \pm 0.05$ | $0.499 \pm 0.01$ |
| 3380 | $2.406 \pm 0.08$ | — | $1.787 \pm 0.05$ | $0.533 \pm 0.02$ | $2.275 \pm 0.05$ | $0.500 \pm 0.01$ |
| 3568 | $2.397 \pm 0.08$ | — | $1.786 \pm 0.05$ | $0.534 \pm 0.02$ | $2.268 \pm 0.05$ | $0.501 \pm 0.01$ |
| 3765 | $2.397 \pm 0.08$ | — | $1.784 \pm 0.05$ | $0.534 \pm 0.03$ | $2.262 \pm 0.05$ | $0.501 \pm 0.01$ |
| 3973 | $2.397 \pm 0.08$ | — | $1.783 \pm 0.05$ | $0.535 \pm 0.03$ | $2.256 \pm 0.05$ | $0.502 \pm 0.01$ |
| 4191 | $2.397 \pm 0.08$ | — | $1.781 \pm 0.05$ | $0.535 \pm 0.03$ | $2.251 \pm 0.05$ | $0.503 \pm 0.01$ |
| 4420 | $2.388 \pm 0.07$ | — | $1.780 \pm 0.05$ | $0.536 \pm 0.03$ | $2.246 \pm 0.05$ | $0.503 \pm 0.01$ |
| 4661 | $2.388 \pm 0.07$ | — | $1.779 \pm 0.05$ | $0.537 \pm 0.03$ | $2.240 \pm 0.05$ | $0.504 \pm 0.01$ |
| 4914 | $2.388 \pm 0.07$ | — | $1.778 \pm 0.05$ | $0.537 \pm 0.03$ | $2.235 \pm 0.05$ | $0.504 \pm 0.01$ |
| 5179 | $2.380 \pm 0.07$ | — | $1.776 \pm 0.05$ | $0.538 \pm 0.03$ | $2.230 \pm 0.05$ | $0.505 \pm 0.01$ |
| 5457 | $2.380 \pm 0.07$ | — | $1.776 \pm 0.05$ | $0.538 \pm 0.03$ | $2.226 \pm 0.05$ | $0.505 \pm 0.01$ |
| 5749 | $2.380 \pm 0.07$ | — | $1.774 \pm 0.05$ | $0.539 \pm 0.03$ | $2.222 \pm 0.05$ | $0.506 \pm 0.01$ |
| 6054 | $2.372 \pm 0.07$ | — | $1.774 \pm 0.05$ | $0.539 \pm 0.03$ | $2.218 \pm 0.05$ | $0.506 \pm 0.01$ |
| 6374 | $2.372 \pm 0.07$ | — | $1.772 \pm 0.05$ | $0.540 \pm 0.03$ | $2.215 \pm 0.05$ | $0.507 \pm 0.01$ |
| 6710 | $2.372 \pm 0.07$ | — | $1.771 \pm 0.05$ | $0.540 \pm 0.03$ | $2.210 \pm 0.05$ | $0.507 \pm 0.01$ |
| 7061 | $2.365 \pm 0.07$ | — | $1.771 \pm 0.05$ | $0.541 \pm 0.03$ | $2.207 \pm 0.05$ | $0.508 \pm 0.01$ |
| 7429 | $2.365 \pm 0.07$ | — | $1.770 \pm 0.05$ | $0.541 \pm 0.03$ | $2.203 \pm 0.05$ | $0.508 \pm 0.01$ |
| 7814 | $2.365 \pm 0.07$ | — | $1.769 \pm 0.05$ | $0.541 \pm 0.03$ | $2.200 \pm 0.05$ | $0.508 \pm 0.01$ |
| 8217 | $2.359 \pm 0.07$ | — | $1.768 \pm 0.05$ | $0.542 \pm 0.03$ | $2.196 \pm 0.05$ | $0.509 \pm 0.01$ |
| 8639 | $2.359 \pm 0.07$ | — | $1.767 \pm 0.05$ | $0.542 \pm 0.03$ | $2.193 \pm 0.05$ | $0.509 \pm 0.01$ |
| 9080 | $2.359 \pm 0.07$ | — | $1.766 \pm 0.05$ | $0.543 \pm 0.03$ | $2.189 \pm 0.05$ | $0.510 \pm 0.01$ |
| 9541 | $2.353 \pm 0.07$ | — | $1.756 \pm 0.05$ | $0.543 \pm 0.03$ | $2.186 \pm 0.05$ | $0.510 \pm 0.01$ |
| 10023 | $2.353 \pm 0.07$ | — | $1.765 \pm 0.05$ | $0.543 \pm 0.03$ | $2.183 \pm 0.05$ | $0.510 \pm 0.01$ |
| 10527 | $2.353 \pm 0.07$ | — | $1.764 \pm 0.05$ | $0.544 \pm 0.03$ | $2.179 \pm 0.05$ | $0.511 \pm 0.01$ |
| 11054 | $2.348 \pm 0.07$ | — | $1.763 \pm 0.05$ | $0.544 \pm 0.03$ | $2.176 \pm 0.05$ | $0.511 \pm 0.01$ |
| 11605 | $2.348 \pm 0.07$ | — | $1.762 \pm 0.05$ | $0.545 \pm 0.03$ | $2.173 \pm 0.05$ | $0.511 \pm 0.01$ |
| 12180 | $2.348 \pm 0.07$ | — | $1.762 \pm 0.05$ | $0.545 \pm 0.03$ | $2.170 \pm 0.06$ | $0.512 \pm 0.01$ |
| 12781 | $2.343 \pm 0.07$ | — | $1.761 \pm 0.05$ | $0.545 \pm 0.03$ | $2.167 \pm 0.06$ | $0.512 \pm 0.01$ |
| 13409 | $2.343 \pm 0.07$ | — | $1.760 \pm 0.05$ | $0.545 \pm 0.03$ | $2.165 \pm 0.06$ | $0.512 \pm 0.01$ |
| 14064 | $2.343 \pm 0.07$ | — | $1.760 \pm 0.05$ | $0.546 \pm 0.03$ | $2.162 \pm 0.06$ | $0.513 \pm 0.01$ |
| 14747 | $2.338 \pm 0.07$ | — | $1.759 \pm 0.05$ | $0.546 \pm 0.03$ | $2.160 \pm 0.06$ | $0.513 \pm 0.01$ |
| 15459 | $2.338 \pm 0.07$ | — | $1.758 \pm 0.05$ | $0.546 \pm 0.03$ | $2.158 \pm 0.06$ | $0.513 \pm 0.01$ |
| 16201 | $2.338 \pm 0.07$ | — | $1.758 \pm 0.05$ | $0.547 \pm 0.03$ | $2.155 \pm 0.06$ | $0.514 \pm 0.01$ |
| 16975 | $2.334 \pm 0.07$ | — | $1.757 \pm 0.05$ | $0.547 \pm 0.03$ | $2.153 \pm 0.06$ | $0.514 \pm 0.01$ |
| 17781 | $2.334 \pm 0.07$ | — | $1.757 \pm 0.05$ | $0.547 \pm 0.03$ | $2.151 \pm 0.06$ | $0.514 \pm 0.01$ |
| 18621 | $2.334 \pm 0.07$ | — | $1.756 \pm 0.05$ | $0.547 \pm 0.03$ | $2.149 \pm 0.06$ | $0.514 \pm 0.01$ |
| 19596 | $2.330 \pm 0.07$ | — | $1.756 \pm 0.05$ | $0.548 \pm 0.03$ | $2.147 \pm 0.06$ | $0.514 \pm 0.01$ |
| 20607 | $2.330 \pm 0.07$ | — | $1.755 \pm 0.05$ | $0.548 \pm 0.03$ | $2.145 \pm 0.06$ | $0.515 \pm 0.01$ |
| 21656 | $2.330 \pm 0.07$ | — | $1.755 \pm 0.05$ | $0.548 \pm 0.03$ | $2.143 \pm 0.06$ | $0.515 \pm 0.01$ |
| 22743 | $2.326 \pm 0.07$ | — | $1.754 \pm 0.05$ | $0.548 \pm 0.03$ | $2.142 \pm 0.06$ | $0.515 \pm 0.01$ |
| 23870 | $2.326 \pm 0.07$ | — | $1.754 \pm 0.05$ | $0.549 \pm 0.03$ | $2.140 \pm 0.06$ | $0.515 \pm 0.01$ |
| 25038 | $2.326 \pm 0.07$ | — | $1.753 \pm 0.05$ | $0.549 \pm 0.03$ | $2.138 \pm 0.06$ | $0.516 \pm 0.01$ |
| 26249 | $2.322 \pm 0.07$ | — | $1.753 \pm 0.06$ | $0.549 \pm 0.03$ | $2.137 \pm 0.06$ | $0.516 \pm 0.01$ |
| 27504 | $2.322 \pm 0.07$ | — | $1.753 \pm 0.06$ | $0.549 \pm 0.03$ | $2.135 \pm 0.06$ | $0.516 \pm 0.01$ |
| 28805 | $2.322 \pm 0.07$ | — | $1.752 \pm 0.06$ | $0.549 \pm 0.03$ | $2.134 \pm 0.06$ | $0.516 \pm 0.01$ |

*Continued on next page*

| $n_{\text{train}}$ | GAT | | GCN | | GraphSAGE | |
|---|---|---|---|---|---|---|
| | Test Loss | Test Acc | Test Loss | Test Acc | Test Loss | Test Acc |
| 30153 | $2.319 \pm 0.07$ | — | $1.752 \pm 0.06$ | $0.550 \pm 0.03$ | $2.133 \pm 0.06$ | $0.516 \pm 0.01$ |
| 31549 | $2.319 \pm 0.07$ | — | $1.751 \pm 0.06$ | $0.550 \pm 0.03$ | $2.132 \pm 0.06$ | $0.516 \pm 0.01$ |
| 32995 | $2.319 \pm 0.07$ | — | $1.751 \pm 0.06$ | $0.550 \pm 0.03$ | $2.130 \pm 0.06$ | $0.517 \pm 0.01$ |
| 33201 | $2.319 \pm 0.07$ | — | $1.751 \pm 0.06$ | $0.550 \pm 0.03$ | $2.130 \pm 0.06$ | $0.517 \pm 0.01$ |

**Note on GAT Results.** The original GAT logs for `Reddit_50k` were not recoverable. We have relaunched all training runs and will update the tables and figures prior to the Dec 3 deadline.

Table 6: Test loss and accuracy (mean $\pm$ std over 20 seeds) for `WorstCase_Bottleneck_20k`.

| $n_{\text{train}}$ | GAT | | GCN | | GraphSAGE | |
|---|---|---|---|---|---|---|
| | Test Loss | Test Acc | Test Loss | Test Acc | Test Loss | Test Acc |
| 300 | $3.208 \pm 0.07$ | $0.406 \pm 0.00$ | $2.077 \pm 0.05$ | $0.387 \pm 0.00$ | $2.511 \pm 0.04$ | $0.395 \pm 0.01$ |
| 318 | $3.117 \pm 0.08$ | $0.401 \pm 0.00$ | $2.122 \pm 0.06$ | $0.414 \pm 0.00$ | $2.566 \pm 0.04$ | $0.417 \pm 0.01$ |
| 338 | $2.926 \pm 0.05$ | $0.404 \pm 0.01$ | $2.111 \pm 0.06$ | $0.405 \pm 0.01$ | $2.438 \pm 0.04$ | $0.419 \pm 0.00$ |
| 360 | $2.739 \pm 0.05$ | $0.408 \pm 0.00$ | $2.072 \pm 0.06$ | $0.412 \pm 0.01$ | $2.409 \pm 0.05$ | $0.424 \pm 0.01$ |
| 382 | $2.694 \pm 0.05$ | $0.408 \pm 0.00$ | $2.065 \pm 0.06$ | $0.417 \pm 0.01$ | $2.340 \pm 0.04$ | $0.426 \pm 0.01$ |
| 407 | $2.546 \pm 0.05$ | $0.408 \pm 0.00$ | $1.911 \pm 0.05$ | $0.411 \pm 0.01$ | $2.289 \pm 0.04$ | $0.425 \pm 0.01$ |
| 432 | $2.452 \pm 0.04$ | $0.408 \pm 0.00$ | $1.837 \pm 0.05$ | $0.416 \pm 0.01$ | $2.246 \pm 0.04$ | $0.426 \pm 0.00$ |
| 459 | $2.383 \pm 0.04$ | $0.410 \pm 0.00$ | $1.786 \pm 0.05$ | $0.414 \pm 0.01$ | $2.177 \pm 0.04$ | $0.430 \pm 0.01$ |
| 488 | $2.351 \pm 0.04$ | $0.411 \pm 0.01$ | $1.731 \pm 0.05$ | $0.421 \pm 0.01$ | $2.116 \pm 0.04$ | $0.434 \pm 0.01$ |
| 519 | $2.185 \pm 0.04$ | $0.412 \pm 0.00$ | $1.715 \pm 0.05$ | $0.426 \pm 0.01$ | $2.052 \pm 0.04$ | $0.437 \pm 0.01$ |
| 552 | $1.990 \pm 0.03$ | $0.415 \pm 0.00$ | $1.663 \pm 0.05$ | $0.432 \pm 0.01$ | $2.030 \pm 0.04$ | $0.440 \pm 0.01$ |
| 587 | $1.887 \pm 0.03$ | $0.415 \pm 0.00$ | $1.651 \pm 0.05$ | $0.435 \pm 0.00$ | $1.986 \pm 0.03$ | $0.443 \pm 0.00$ |
| 624 | $1.832 \pm 0.04$ | $0.412 \pm 0.00$ | $1.637 \pm 0.05$ | $0.435 \pm 0.00$ | $1.980 \pm 0.04$ | $0.443 \pm 0.01$ |
| 663 | $1.774 \pm 0.04$ | $0.415 \pm 0.00$ | $1.613 \pm 0.04$ | $0.442 \pm 0.01$ | $1.960 \pm 0.04$ | $0.441 \pm 0.01$ |
| 705 | $1.763 \pm 0.07$ | $0.418 \pm 0.01$ | $1.579 \pm 0.04$ | $0.442 \pm 0.00$ | $1.937 \pm 0.03$ | $0.443 \pm 0.00$ |
| 749 | $1.806 \pm 0.09$ | $0.426 \pm 0.01$ | $1.530 \pm 0.04$ | $0.449 \pm 0.00$ | $1.901 \pm 0.03$ | $0.448 \pm 0.00$ |
| 796 | $1.749 \pm 0.03$ | $0.430 \pm 0.00$ | $1.505 \pm 0.04$ | $0.452 \pm 0.01$ | $1.870 \pm 0.03$ | $0.446 \pm 0.00$ |
| 846 | $1.689 \pm 0.03$ | $0.438 \pm 0.01$ | $1.485 \pm 0.04$ | $0.453 \pm 0.00$ | $1.824 \pm 0.03$ | $0.448 \pm 0.00$ |
| 900 | $1.627 \pm 0.03$ | $0.438 \pm 0.00$ | $1.437 \pm 0.04$ | $0.456 \pm 0.00$ | $1.795 \pm 0.03$ | $0.449 \pm 0.00$ |
| 956 | $1.586 \pm 0.03$ | $0.441 \pm 0.00$ | $1.407 \pm 0.04$ | $0.459 \pm 0.01$ | $1.763 \pm 0.03$ | $0.450 \pm 0.01$ |
| 1017 | $1.545 \pm 0.03$ | $0.440 \pm 0.01$ | $1.389 \pm 0.04$ | $0.462 \pm 0.00$ | $1.704 \pm 0.03$ | $0.455 \pm 0.01$ |
| 1081 | $1.531 \pm 0.02$ | $0.440 \pm 0.01$ | $1.377 \pm 0.03$ | $0.463 \pm 0.00$ | $1.652 \pm 0.02$ | $0.457 \pm 0.01$ |
| 1149 | $1.502 \pm 0.02$ | $0.437 \pm 0.00$ | $1.349 \pm 0.03$ | $0.468 \pm 0.00$ | $1.613 \pm 0.02$ | $0.462 \pm 0.00$ |
| 1221 | $1.466 \pm 0.02$ | $0.435 \pm 0.00$ | $1.323 \pm 0.03$ | $0.468 \pm 0.00$ | $1.573 \pm 0.02$ | $0.462 \pm 0.01$ |
| 1298 | $1.425 \pm 0.02$ | $0.434 \pm 0.00$ | $1.310 \pm 0.03$ | $0.470 \pm 0.00$ | $1.554 \pm 0.02$ | $0.460 \pm 0.00$ |
| 1379 | $1.389 \pm 0.01$ | $0.437 \pm 0.01$ | $1.294 \pm 0.03$ | $0.471 \pm 0.00$ | $1.549 \pm 0.02$ | $0.460 \pm 0.00$ |
| 1465 | $1.362 \pm 0.02$ | $0.436 \pm 0.00$ | $1.276 \pm 0.02$ | $0.472 \pm 0.00$ | $1.519 \pm 0.02$ | $0.461 \pm 0.01$ |
| 1556 | $1.335 \pm 0.01$ | $0.435 \pm 0.00$ | $1.258 \pm 0.02$ | $0.475 \pm 0.00$ | $1.494 \pm 0.02$ | $0.461 \pm 0.01$ |
| 1653 | $1.303 \pm 0.01$ | $0.438 \pm 0.01$ | $1.240 \pm 0.02$ | $0.478 \pm 0.00$ | $1.469 \pm 0.02$ | $0.466 \pm 0.00$ |
| 1755 | $1.278 \pm 0.01$ | $0.438 \pm 0.01$ | $1.224 \pm 0.02$ | $0.480 \pm 0.00$ | $1.444 \pm 0.02$ | $0.468 \pm 0.01$ |
| 1863 | $1.257 \pm 0.01$ | $0.440 \pm 0.01$ | $1.211 \pm 0.02$ | $0.482 \pm 0.00$ | $1.430 \pm 0.02$ | $0.470 \pm 0.01$ |
| 1977 | $1.240 \pm 0.01$ | $0.441 \pm 0.00$ | $1.198 \pm 0.02$ | $0.483 \pm 0.00$ | $1.402 \pm 0.01$ | $0.473 \pm 0.01$ |
| 2098 | $1.225 \pm 0.01$ | $0.441 \pm 0.00$ | $1.186 \pm 0.02$ | $0.484 \pm 0.00$ | $1.382 \pm 0.01$ | $0.474 \pm 0.01$ |
| 2226 | $1.213 \pm 0.01$ | $0.443 \pm 0.00$ | $1.177 \pm 0.02$ | $0.487 \pm 0.00$ | $1.365 \pm 0.01$ | $0.475 \pm 0.01$ |
| 2362 | $1.201 \pm 0.01$ | $0.444 \pm 0.00$ | $1.168 \pm 0.02$ | $0.489 \pm 0.00$ | $1.351 \pm 0.01$ | $0.477 \pm 0.01$ |
| 2507 | $1.190 \pm 0.01$ | $0.445 \pm 0.00$ | $1.159 \pm 0.02$ | $0.490 \pm 0.00$ | $1.336 \pm 0.01$ | $0.480 \pm 0.01$ |
| 2661 | $1.181 \pm 0.01$ | $0.447 \pm 0.00$ | $1.152 \pm 0.02$ | $0.491 \pm 0.00$ | $1.316 \pm 0.01$ | $0.483 \pm 0.01$ |
| 2824 | $1.172 \pm 0.01$ | $0.448 \pm 0.00$ | $1.145 \pm 0.02$ | $0.492 \pm 0.00$ | $1.304 \pm 0.01$ | $0.485 \pm 0.01$ |
| 2997 | $1.165 \pm 0.01$ | $0.449 \pm 0.00$ | $1.138 \pm 0.02$ | $0.494 \pm 0.00$ | $1.290 \pm 0.01$ | $0.485 \pm 0.01$ |
| 3180 | $1.158 \pm 0.01$ | $0.450 \pm 0.00$ | $1.132 \pm 0.02$ | $0.495 \pm 0.00$ | $1.278 \pm 0.01$ | $0.487 \pm 0.01$ |
| 3374 | $1.153 \pm 0.01$ | $0.451 \pm 0.00$ | $1.127 \pm 0.02$ | $0.497 \pm 0.00$ | $1.267 \pm 0.01$ | $0.489 \pm 0.01$ |
| 3579 | $1.148 \pm 0.01$ | $0.452 \pm 0.00$ | $1.122 \pm 0.02$ | $0.498 \pm 0.00$ | $1.256 \pm 0.01$ | $0.490 \pm 0.01$ |
| 3796 | $1.143 \pm 0.01$ | $0.453 \pm 0.00$ | $1.117 \pm 0.02$ | $0.499 \pm 0.00$ | $1.248 \pm 0.01$ | $0.490 \pm 0.01$ |
| 4025 | $1.139 \pm 0.01$ | $0.454 \pm 0.00$ | $1.112 \pm 0.02$ | $0.499 \pm 0.01$ | $1.240 \pm 0.01$ | $0.492 \pm 0.01$ |
| 4267 | $1.135 \pm 0.01$ | $0.454 \pm 0.00$ | $1.108 \pm 0.02$ | $0.500 \pm 0.01$ | $1.232 \pm 0.01$ | $0.493 \pm 0.01$ |

*Continued on next page*

| $n_{\text{train}}$ | GAT | | GCN | | GraphSAGE | |
|---|---|---|---|---|---|---|
| | Test Loss | Test Acc | Test Loss | Test Acc | Test Loss | Test Acc |
| 4523 | $1.131 \pm 0.01$ | $0.455 \pm 0.00$ | $1.105 \pm 0.02$ | $0.501 \pm 0.01$ | $1.227 \pm 0.01$ | $0.493 \pm 0.01$ |
| 4793 | $1.128 \pm 0.01$ | $0.456 \pm 0.00$ | $1.101 \pm 0.02$ | $0.502 \pm 0.01$ | $1.219 \pm 0.01$ | $0.494 \pm 0.01$ |
| 5078 | $1.125 \pm 0.01$ | $0.456 \pm 0.00$ | $1.097 \pm 0.01$ | $0.502 \pm 0.01$ | $1.214 \pm 0.01$ | $0.495 \pm 0.01$ |
| 5379 | $1.122 \pm 0.01$ | $0.457 \pm 0.00$ | $1.094 \pm 0.01$ | $0.503 \pm 0.01$ | $1.210 \pm 0.01$ | $0.496 \pm 0.01$ |
| 5696 | $1.119 \pm 0.01$ | $0.458 \pm 0.00$ | $1.091 \pm 0.01$ | $0.504 \pm 0.01$ | $1.204 \pm 0.01$ | $0.497 \pm 0.01$ |
| 6030 | $1.116 \pm 0.01$ | $0.458 \pm 0.00$ | $1.088 \pm 0.01$ | $0.504 \pm 0.01$ | $1.200 \pm 0.01$ | $0.497 \pm 0.01$ |
| 6381 | $1.114 \pm 0.01$ | $0.459 \pm 0.00$ | $1.086 \pm 0.01$ | $0.504 \pm 0.01$ | $1.196 \pm 0.01$ | $0.498 \pm 0.01$ |
| 6751 | $1.112 \pm 0.01$ | $0.459 \pm 0.00$ | $1.083 \pm 0.01$ | $0.505 \pm 0.01$ | $1.192 \pm 0.01$ | $0.498 \pm 0.01$ |
| 7139 | $1.109 \pm 0.01$ | $0.460 \pm 0.00$ | $1.081 \pm 0.01$ | $0.505 \pm 0.01$ | $1.189 \pm 0.01$ | $0.499 \pm 0.01$ |
| 7547 | $1.107 \pm 0.01$ | $0.460 \pm 0.00$ | $1.079 \pm 0.01$ | $0.506 \pm 0.01$ | $1.186 \pm 0.01$ | $0.499 \pm 0.01$ |
| 7975 | $1.105 \pm 0.01$ | $0.460 \pm 0.00$ | $1.077 \pm 0.01$ | $0.506 \pm 0.01$ | $1.183 \pm 0.01$ | $0.500 \pm 0.01$ |
| 8424 | $1.104 \pm 0.01$ | $0.461 \pm 0.00$ | $1.075 \pm 0.01$ | $0.506 \pm 0.01$ | $1.180 \pm 0.01$ | $0.500 \pm 0.01$ |
| 8895 | $1.102 \pm 0.01$ | $0.461 \pm 0.01$ | $1.073 \pm 0.01$ | $0.507 \pm 0.01$ | $1.178 \pm 0.01$ | $0.501 \pm 0.01$ |
| 9389 | $1.100 \pm 0.01$ | $0.461 \pm 0.00$ | $1.071 \pm 0.01$ | $0.507 \pm 0.01$ | $1.175 \pm 0.01$ | $0.501 \pm 0.01$ |
| 11000 | $1.086 \pm 0.01$ | $0.471 \pm 0.00$ | $1.021 \pm 0.00$ | $0.482 \pm 0.00$ | $1.060 \pm 0.00$ | $0.476 \pm 0.00$ |

## R  SYNTHETIC EXPERIMENTS VERIFYING THE MINIMAX SCALING LAW

This appendix provides controlled synthetic experiments validating the worst–case minimax rate established in Theorem 1. We directly instantiate the least–favorable construction:

$$\text{Err}(n, d) \;=\; C\sqrt{\frac{\log d}{n}}\,(1 + \eta),$$

with a small multiplicative noise term $\eta$ to mimic empirical variability.

The main text (Figure 1) uses the collapse test—the most sensitive diagnostic—to show that only the normalization $\sqrt{\log d/n}$ removes both $n$- and $d$-dependence. For completeness, this appendix includes two orthogonal sanity-checks:

1. **Error vs. $n$ for multiple $d$**  Confirms the slope in $n$ is exactly $n^{-1/2}$ for every fixed $d$, and the only effect of increasing $d$ is a vertical shift proportional to $\sqrt{\log d}$.

2. **Error vs. $d$ for multiple $n$**  Confirms the error grows in $d$ exactly at the rate $\sqrt{\log d}$, and that larger $n$ only rescales curves downward by the factor $1/\sqrt{n}$.

Together, these experiments verify both axes of the minimax law and complement the collapse-based evidence shown in the main text.

**Interpretation of Figure 8.**  All curves follow the predicted $n^{-1/2}$ slope on the log–log scale. Increasing $d$ produces a parallel vertical shift exactly equal to $\sqrt{\log d}$, with no change in slope. This directly verifies the $n$-axis behavior of the minimax rate.

**Interpretation of Figure 9.**  For each fixed $n$, the error grows as $\sqrt{\log d}$, producing smooth monotone curves in $d$. The family of curves for increasing $n$ differ only by the $1/\sqrt{n}$ scaling factor. This verifies the $d$-axis behavior of the minimax rate.

In combination with the collapse experiment in the main text, these two synthetic plots provide full empirical confirmation of the minimax lower bound in Theorem 1.

## S  THE USE OF LARGE LANGUAGE MODELS (LLMS)

In preparing this manuscript, we used Large Language Models (LLMs) solely as general-purpose assistive tools for grammar checking, language polishing, and improving clarity of exposition. LLMs were not used for research ideation, theoretical development, experiment design, or analysis, and they did not contribute any scientific content. The authors take full responsibility for the contents of the paper, including any parts where LLMs were used to improve writing style.

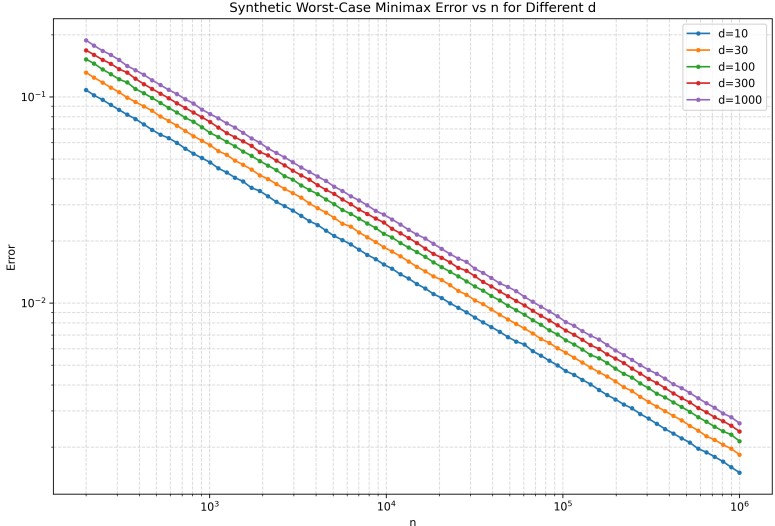

Figure 8: Error vs. $n$ for different $d$.

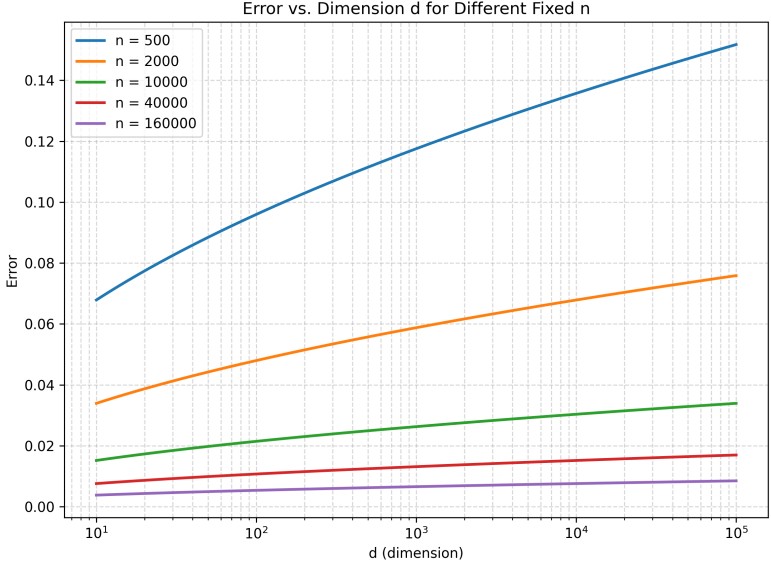

Figure 9: Error vs. $d$ for different fixed $n$.

